# The Skp1-Cullin1-FBXO1 complex is a pleiotropic regulator required for the formation of gametes and motile forms in *Plasmodium berghei*

**Ravish Rashpa** [1] ✉, **Natacha Klages**[1], **Domitille Schvartz**[2], **Carla Pasquarello**[2] & **Mathieu Brochet** [1] ✉

Malaria-causing parasites of the *Plasmodium* genus undergo multiple developmental phases in the human and the mosquito hosts, regulated by various post-translational modifications. While ubiquitination by multi-component E3 ligases is key to regulate a wide range of cellular processes in eukaryotes, little is known about its role in *Plasmodium*. Here we show that *Plasmodium berghei* expresses a conserved SKP1/Cullin1/FBXO1 (SCF[FBXO1]) complex showing tightly regulated expression and localisation across multiple developmental stages. It is key to cell division for nuclear segregation during schizogony and centrosome partitioning during microgametogenesis. It is additionally required for parasite-specific processes including gamete egress from the host erythrocyte, as well as integrity of the apical and the inner membrane complexes (IMC) in merozoite and ookinete, two structures essential for the dissemination of these motile stages. Ubiquitinomic surveys reveal a large set of proteins ubiquitinated in a FBXO1-dependent manner including proteins important for egress and IMC organisation. We additionally demonstrate an interplay between FBXO1-dependent ubiquitination and phosphorylation via calcium-dependent protein kinase 1. Altogether we show that *Plasmodium* SCF[FBXO1] plays conserved roles in cell division and is also important for parasite-specific processes in the mammalian and mosquito hosts.

Malaria is caused by unicellular parasites of the genus *Plasmodium*, which are transmitted by mosquitoes. Clinical symptoms associated with malaria are caused by the asexual replication of erythrocytic blood stages with waves of fever arising from the synchronised egress of merozoites from erythrocytes. Merozoite formation involves the assembly of two interconnected structures important for cell scaffolding and motility: the inner membrane complex (IMC) and the apical complex. The IMC lies below the plasma membrane and consists of flattened membrane sacs called alveoli[1]. During the merozoite formation, the IMC functions as an important scaffolding compartment

following rounds of nuclear division[2]. In mature merozoites, it also serves as an anchor to the actomyosin motor apparatus that is required for motility and host cell invasion. The cytoplasmic leaflet of the IMC is associated with an intermediate-filament-like cytoskeletal network and subpellicular microtubules, which confer mechanical strength to the merozoite. The apical complex includes specialised secretory organelles releasing factors necessary for egress, motility and invasion[3] as well as the apical polar ring (APR) from which emanate the subpellicular microtubules. Above the APR is the conoid complex that is also critical for motility and host cell invasion[4].

[1]University of Geneva, Faculty of Medicine, Department of Microbiology and Molecular Medicine, Geneva, Switzerland. [2]University of Geneva, Faculty of Medicine, Proteomics Core Facility, Geneva, Switzerland. ✉e-mail: Ravish.Rashpa@unige.ch; Mathieu.Brochet@unige.ch

Parasite transmission to mosquitoes relies on macro- (female) and micro- (male) gametocytes that differentiate inside erythrocytes. Upon maturation, gametocytes become quiescent, waiting for uptake during a mosquito blood feed to resume their development[5]. Gametocytes resume their development in response to environmental cues, including a small mosquito molecule, xanthurenic acid (XA), a rise in extracellular pH and a concomitant drop in temperature[6]. The Egress of gametes from the host erythrocyte occurs within fifteen minutes of gametocyte activation. During this time, the microgametocyte completes three rounds of genome replication and closed mitosis within a single nucleus, and assembles eight axonemes that lead to the formation of eight flagellated microgametes in a process called exflagellation. Following activation of translationally repressed mRNAs, macrogametes are available for the fertilisation to colonise the mosquito midgut[7]. Within 16 hours of fertilisation, zygotes differentiate into motile ookinetes, which colonise the epithelial monolayer of the mosquito midgut. Each ookinete transforms into an oocyst that undergoes sporogony. Eventually, thousands of sporozoites are released from each cyst and reach the salivary glands of the mosquito. Once released into another human, parasites first replicate in the liver before re-entering the bloodstream.

Transcriptional regulation has been shown to play an important part in coordinating these multiple phases of parasite development[8,9]. However, key aspects of cellular development also require post-translational modifications (PTMs) of proteins[10]. Multiple PTMs have been shown to be important for developmental transitions, including phosphorylation, acetylation, methylation, and lipidation[10–12], but little is known about the requirement of ubiquitination in *Plasmodium*. Mass spectrometry-based approaches identified snapshots of ubiquitinated proteins in *Plasmodium*[13,14]. More recently, 1464 sites in 546 proteins were identified in asexual blood stages with a large increase of ubiquitinated sites associated with merozoite maturation[15]. Interestingly, the ubiquitination of components of the IMC, the apical complex and the actomyosin motor was enriched, suggesting an important role in egress and invasion. Regulation by ubiquitination is well-known to be mediated through the ubiquitin-proteasome system, but ubiquitination was also shown to regulate protein localisation, interaction or enzymatic activity[16]. However, despite the key role of ubiquitination in eukaryotic cells, its functional requirement during the *Plasmodium* lifecycle remains elusive.

Ubiquitin is a 76-residue protein that is usually added through a covalent bond to lysine residues in substrate proteins. Protein ubiquitination is a three-step process during which ubiquitin is first activated by an E1 ubiquitin-activating enzyme, transferred to an E2 ubiquitin-conjugating enzyme and then transferred to a substrate selected by an E3 ubiquitin ligase. E3 ligases are generally multiprotein complexes and are classified into four families: HECT, RING-finger, U-box, and PHD-finger. Among the RING-finger ligases, the cullin-RING E3 ligases (CRLs) represent a diverse group that includes ligases such as the Skp1/Cullin/F-box (SCF) protein complex and the anaphase-promoting complex/cyclosome (APC/C). These CRLs have important roles in the ubiquitination of proteins involved in the cell cycle. Four APC/C components have been identified in the *Plasmodium* genome: APC3, APC10, APC11, as well as the adaptor protein CDC20. APC3 and CDC20 were shown to be important for *Plasmodium* microgametogenesis[17,18]. SCF complexes contain three invariant proteins, RBX1, Cullin and SKP1, in addition to a variable F-box protein. Cullin serves as a scaffold protein which N-terminally binds to SKP1 and C-terminally binds to a RING-finger protein (RING-box protein 1 - RBX1) and an E2 ubiquitin-conjugating enzyme. F-box proteins generally contain a ~50 amino acid F-box domain which functions as a receptor by binding to the SCF complex. F-box domains are commonly found in concert with other protein–protein interaction motifs, such as leucine-rich, WD repeat and ankyrin repeats, which mediate interactions with their substrate-specific interaction motif. Genome mining in

*Plasmodium* identified SKP1, Cullin1, and RBX1 together with two F-box domain-containing proteins[19]. However, their roles during the *Plasmodium* development remain unknown.

In this work, we first set out to analyse the repertoire and function of SCF complexes in *Plasmodium*. We reasoned that sexual stages would represent an ideal starting point given the high synchronicity of cell cycle progression and development, that has been key to elucidate cell cycle structures and multiple phospho-dependent signalling pathways[20,21]. We first show that gametogenesis and ookinete development require the ubiquitin-proteasome system and we identify in developing gametocytes 2183 ubiquitinated lysine residues mapping on 519 proteins. Immunoprecipitations followed by mass spectrometry confirmed the expression of a stable SCF complex with multiple adaptors, including FBXO1. Localisation of proteins by ultrastructure expansion microscopy (U-ExM) indicates that SKP1 and FBXO1 show overlapping cellular distribution during sexual development. Functional analysis reveals that centrosome homoeostasis, timely gamete egress from the host erythrocyte and integrity of the pellicle and subpellicular microtubules in developing ookinetes all require SCF^FBXO1. These two last phenotypes are overlapping with those described for CDPK1-depleted gametocytes, a kinase that we also find enriched in FBXO1 immunoprecipitates. A subset of 32 proteins is inversely regulated in CDPK1- and FBXO1-dependent ubiquitomes highlighting a possible interplay between ubiquitination and phosphorylation. Consistent with this idea, we further demonstrate that CDPK1 is required for FBXO1 proteostasis in gametocytes. Furthermore, mutating a single ubiquitinated lysine residue on CDPK1 phenocopies *cdpk1* deletion, suggesting a bidirectional interplay between ubiquitination and phosphorylation. Finally, we show an important requirement of FBXO1 during late schizogony for nuclear segregation, formation of the apical complex and merozoite segmentation.

## Results

### Dynamic ubiquitination and proteasome degradation during *P. berghei* sexual development

We first aimed at estimating the global extent of ubiquitination dynamics during *P. berghei* sexual development by monitoring ubiquitin-protein conjugates by western blot with anti-ubiquitin-K48 antibodies (Fig. 1a). As parasites progress through gametogenesis, the ubiquitination signal increased up to eight minutes post-activation but dramatically decreased 15 minutes post-activation (Fig. 1a). To test whether this decrease was linked to proteasome degradation, we treated cells with either 1 μM MG132 (Fig. 1a) or 25 μM bortezomib (Fig. S1a) two molecules that were shown to target the catalytic site of the 26 S proteasome in other eukaryotic systems. These treatments increased ubiquitin signals suggesting that the ubiquitin-proteasome system is active during gametogenesis. Consistent with this, the same proteasome inhibitors were shown to impact gametogenesis in *Plasmodium*[22], which we confirmed here (Fig. 1b). We additionally investigated the effect of 1 μM MG132 on ookinete development by treating parasites 1 h post-activation. This arrested ookinete development at a point reminiscent of the "retort" stage of normal ookinete development (Fig. 1c, d and Fig. S1b). Further investigation of the ookinete microtubule cytoskeleton by U-ExM indicated that treated parasites formed subpellicular microtubules (Fig. S1c).

### Global identification of ubiquitination events in developing gametocytes

To identify endogenous ubiquitination sites in developing gametocytes, we applied a ubiquitin remnant immunoaffinity profiling technique[23]. Trypsin digestion of proteins results in the cleavage at arginine and lysine residues, where the C-terminal Gly-Gly dipeptide of ubiquitin is still attached to the ubiquitinated lysine residue of substrate proteins. Using commercial antibodies that recognise this Gly-Gly remnant, peptides were enriched prior to LC-MS/MS.

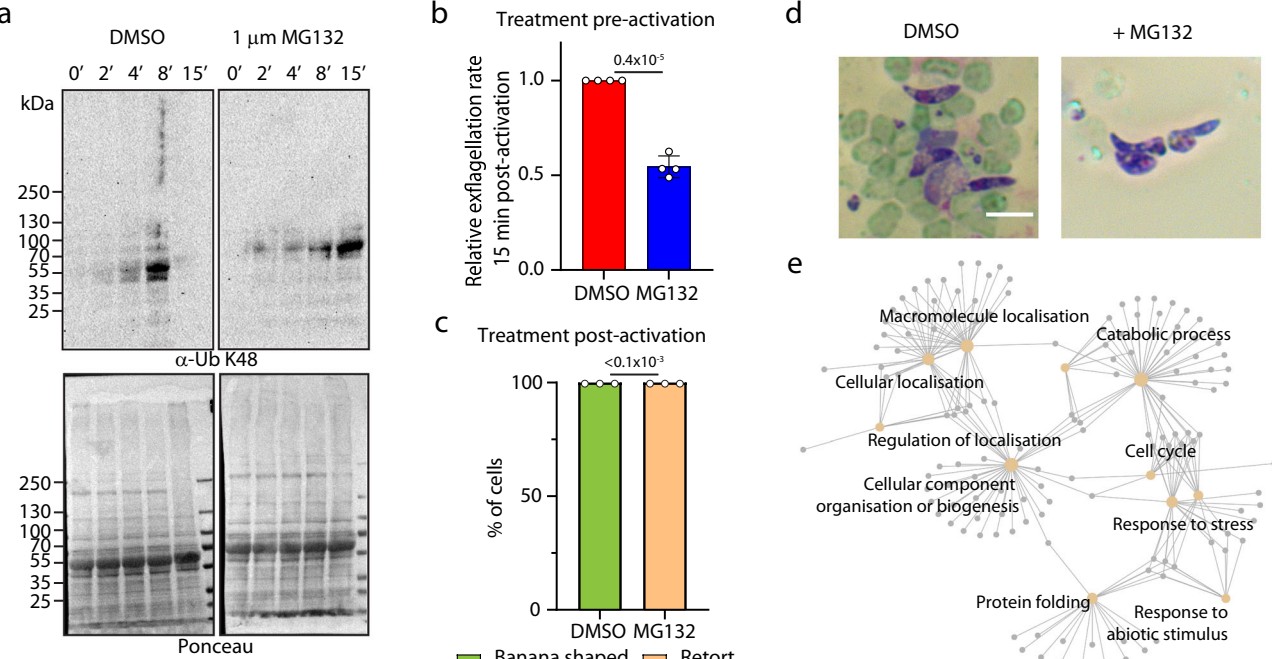

**Fig. 1 | Dynamic ubiquitination during *P. berghei* sexual development.**
**a** Western blot analysis with an antibody against ubiquitin-K48 (α-Ub K48) of gametocyte lysates collected at different time points during the first 15 min following activation by xanthurenic acid. The Ponceau staining serves as a loading control. **b** Treatment pre-activation of gametocytes with 1 μM MG132 leads to reduced exflagellation (error bars show standard deviation from the mean; replicates from four independent infections; unpaired two-tailed *t*-test). **c** Treatment of

gametocytes with 1 μM MG132 one hour post-activation leads to altered ookinete formation (replicates from three independent infections; unpaired two-tailed *t*-test). **d** Representative images of Giemsa-stained ookinete showing DMSO banana-shaped and MG132-treated retort ookinetes. Scale bar = 5 μm. **e** GO term enrichment analysis of ubiquitinated proteins in non-activated and 4 min activated WT gametocytes. Source data are provided as a Source Data file.

Data-independent acquisition (DIA)-MS method with a neural network-based data processing were combined to quantify the modified proteomes. Samples were collected in non-activated gametocytes and four to six minutes post-activation, a time at which our western blot analysis suggested an increase in global ubiquitination levels (Fig. 1a). To limit degradation of ubiquitinated proteins, gametocytes were treated with 1 μM MG132 30 min pre-activation. In parallel, we analysed the full proteome in the absence of MG132. We identified 2565 proteins (Supplementary Data 1) and no significant major changes could be observed between non-activated and early-activated gametocytes.

In the Gly-Gly enriched fraction, we identified 2183 ubiquitinated lysine residues mapping on 519 proteins (Supplementary Data 1) with no major pathways enriched between the two time points (Fig. S1d). No strong correlation between protein levels and ubiquitination was detected (Fig. S1e). Analysis of the sequences around the ubiquitination sites identified five enriched motifs representing 744 sites (Fig. S1f). GO term enrichment analyses of ubiquitinated proteins revealed a large cluster of terms associated with protein degradation and ubiquitination. The analysis also suggested potential regulatory roles for ubiquitination, including cell cycle progression and cellular component organisation (Fig. 1e and Supplementary Data 1). Interestingly, a subset of exported proteins was also found to be ubiquitinated, raising the possibility that the ubiquitination of parasite proteins also takes place in the host cell. Altogether these results indicate a large repertoire of ubiquitinated proteins in *Plasmodium* gametocytes suggesting an important regulatory role during sexual development.

### *P. berghei* gametocytes express a conserved SCF complex with multiple putative adaptor proteins

*Plasmodium* genome mining previously identified orthologues of four proteins possibly belonging to SCF complexes:[14,24] Cullin1 (CUL1 - PBANKA_1426500), RBX1 (PBANKA_0806200), SKP1 (PBANKA_

1142900), Nedd8 (PBANKA_1411400), and a second cullin domain-containing protein (CUL2 - PBANKA_1128600). We first set out to identify whether these proteins form a complex in *Plasmodium* using affinity purification of triple HA-tagged proteins. To do so, we generated transgenic *P. berghei* lines expressing triple HA-tagged RBX1, SKP1 and CUL1 (Fig. S2a). Immunoblotting confirmed the expression of fusion proteins in gametocytes with the expected mobility for RBX1-HA and SKP1-HA (Fig. 2a), but no signal above the background could be observed for CUL1-HA (Fig. S2b) from two independent lines, despite correct in-frame integration of the HA tag (see method section). Immunofluorescence assays (IFA) revealed a cell-wide distribution of both RBX1-HA and SKP1-HA in both activated macro- and microgametocytes, while no signal above the detection limit was detected for CUL1-HA gametocytes.

To identify interacting proteins for RBX1-HA and SKP1-HA, we then used affinity purification of triple HA-tagged proteins from synchronised mitotic gametocytes 4–6 min post-activation followed by label-free semiquantitative mass spectrometry (Supplementary Data 2). Supporting the notion of an SKP1/CUL1 complex, CUL1, Nedd8 and polyubiquitin (PolyUb – PBANKA_ 0610300) were enriched proteins co-purifying with RBX1-HA and SKP1-HA (Fig. 2c). Nedd8 shares 60% amino acid sequence identity to ubiquitin and is known to target cullins to activate their respective SCF complex. Among the proteins most enriched in both immunoprecipitates were also two proteins containing either F-box (FBXO1 – PBANKA_1118900) or leucine-rich repeats (FBXL2 or LRR11, PBANKA_0925100) that could represent possible adaptors of the SKP1/CUL1 complex. We also identified the IMC sub-compartment protein 1 (ISP1 - PBANKA_1209400), suggesting that this protein, which is important to define the polarity of ookinetes[25,26], may be regulated by SCF-dependent ubiquitination. A protein of unknown function, PBANKA_1358700 was also enriched in all immunoprecipitates. Interestingly, other proteins only copurified

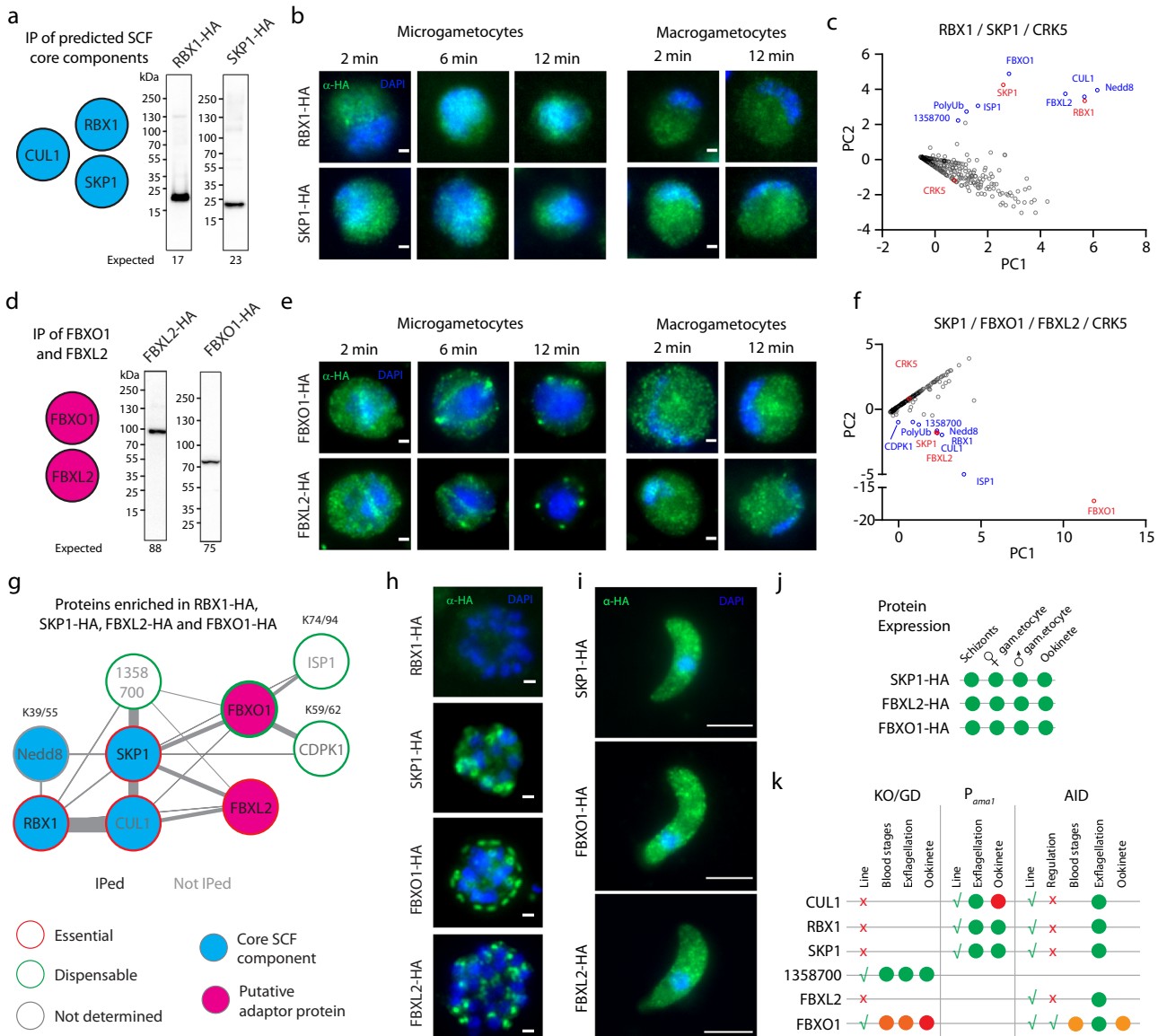

**Fig. 2 | *P. berghei* expresses SCF complexes with multiple requirements during sexual development. a** Western blot analysis of gametocyte lysates from lines expressing endogenously HA-tagged alleles of core SCF components: RBX1-HA and SKP1-HA. **b** Localisation of RBX1-HA and SKP1-HA (green) by widefield immunofluorescence in micro- and macro-gametocytes between two- and 12-min post-activation. DNA is stained with DAPI (blue). Scale bar = 1 μm. **c** emPAI values for proteins co-purifying with RBX1-HA and SKP1-HA following immunoprecipitation, and displayed in first and second principal components (n = 2 biological replicates). Red denotes immunoprecipitated proteins and blue possible components of the SCF complex. **d** Western blot analysis of gametocyte lysates from lines expressing endogenously HA-tagged alleles of FBXO1-HA and FBXL2-HA. **e** Localisation of FBXO1-HA and FBXL2-HA (both green) by widefield immunofluorescence in gametocytes between two- and 12-min post-activation. DNA is stained with DAPI (blue). Scale bar = 1 μm. **f** emPAI values for proteins co-purifying with SKP1-HA, FBXO1-HA and FBXL2 displayed in first and second principal components (n = 2

biological replicates). CRK5-HA was used as a control. Red denotes immunoprecipitated proteins and blue possible components of the SCF complex.
**g** Protein–protein interaction network analysis of immuno-purification enrichment from **c** and **f**. Ubiquitinated lysine residues are indicated. **h, i** Localisation of RBX1-HA, SKP1-HA, FBXO1-HA and FBXL2-HA (all green), by widefield immuno-fluorescence in segmenting schizonts (**h**) and in ookinetes (**i**). DNA is stained with DAPI (blue). Scale bars: schizont = 1 μm, ookinete = 5 μm. **j** Summary of the expression profiles of components of three proteins of the SCF^FBXO1 complex.
**k** Overview of the reverse genetic approaches used to investigate the function of six proteins highlighted in **g**. Red crosses indicate that the line could not be generated, while green ticks denote lines that could be generated. Green dots indicate that no phenotype was observed, while orange and red dots indicate a partial defect or a complete block, respectively. KO knockout, GD gene disruption. Source data are provided as a Source Data file.

with RBX1-HA or SKP1-HA, respectively. These included CUL2 that was only co-immunoprecipitated with RBX1-HA, suggesting a diverse repertoire of CRLs (Supplementary Data 2).

To confirm FBXO1 and FBXL2 are components of SKP1/CUL1 complexes, we generated transgenic *P. berghei* lines expressing endogenously triple HA-tagged alleles of FBXO1 and FBXL2 (Fig. S2a). Immunoblotting confirmed the expression of fusion proteins in

gametocytes with the expected mobility (Fig. 2d). IFA showed a similar cytoplasmic distribution of FBXO1-HA and FBXL2-HA in both macro- and microgametes with slightly brighter staining at structures that likely correspond to mitotic spindles and forming axonemes in the microgametes (Fig. 2e). Immunoprecipitations of FBXO1-HA or FBXL2-HA enriched peptides from SKP1, CUL1, RBX1, PolyUb and Nedd8, while the most enriched protein was ISP1 (Fig. 2f). Interestingly, CDPK1,

a regulator of merozoite formation, gamete egress and ookinete development[27–30], was also found to be slightly enriched in FBXO1-HA immunoprecipitates.

Altogether these results indicate that gametocytes express a conserved SKP1/RBX1/CUL1 complex with at least two possible adaptor proteins, FBXO1 and FBXL2 (Fig. 2g). The enrichment of CDPK1 in FBXO1-HA and SKP1-HA immunoprecipitates additionally suggests a possible interplay between phosphorylation and ubiquitinationto regulate sexual development.

### Components of the SCF^FBXO1 complex are also expressed in schizonts and ookinetes

Previous transcriptomic and proteomic surveys have indicated that RBX1, SKP1, FBXO1 and FBXL2 are also expressed in schizonts and ookinetes[31–33]. We thus took advantage of the respective HA-tagged lines to further investigate their expression and localisation in both stages. While a faint cytoplasmic signal could be observed for RBX1-HA in schizonts, no specific enrichment could be observed at this resolution (Fig. 2h). In contrast, SKP1-HA, FBXO1-HA, and FBXL2-HA showed a pattern that was reminiscent of the dynamic localisation of the glideosome-associated protein 45 (GAP45) during *P. falciparum* late schizogony[34] (Fig. 2h). A signal above the detection limit was also observed for SKP1-HA, FBXO1-HA and FBXL2-HA in ookinetes (Fig. 2i). Altogether, these results show that components of the SCF^FBXO1 complex identified in gametocytes are also expressed in schizonts and ookinetes suggesting multistage requirements for these proteins (Fig. 2j).

### Functional interrogation of SCF^FBXO1 components indicates requirements for asexual blood stages, gametogenesis and ookinete development

To study the function of SCF^FBXO1, we first attempted to knock out *skp1*, *cul1*, and *rbx1*. The two latter genes were previously suggested to be resistant to disruption in a global gene knockout study[35] and we were also unable to obtain KO populations (Fig. 2k) despite four independent attempts. We then tagged endogenous genes with an auxin-inducible degron (AID) coupled to an HA epitope tag (Fig. S2c) that allows degradation of the fusion protein in the presence of auxin in a strain expressing the Tir1 protein[36]. However, no significant degradation of the targeted proteins nor defects in exflagellation could be observed upon auxin addition in non-clonal populations (Fig. 2k and Fig. S2d). This prevented us to further use this system to interrogate the functions of these proteins across multiple stages. To infer potential function in gametocytes, we then opted for stage-specific knockdowns by placing the endogenous *cul1* or *rbx1* genes under the control of the *pbama1* promoter, which is active in schizonts but virtually silent in gametocytes[27]. Clones were readily obtained (Fig. S2e), but none of them showed quantifiable defects in exflagellation (Fig. S2f), possibly due to sufficient expression levels. However, the P_{ama1}CUL1 clone mainly formed retort ookinetes (Fig. S2g), a phenotype that was partially rescued by fertilisation with competent Nek4-KO microgametes[37], suggestive of a role for *cul1* in either sexual lineage.

We then interrogated the requirement for the putative adaptor proteins FBXO1, FBXL2 and the protein of unknown function PBANKA_1358700 that was enriched in SCF^FBXO1 immunoprecipitates. We were able to obtain a KO clonal line for PBANKA_1358700, but no defects in asexual blood stages nor in exflagellation were detected (Fig. S2h, i). No FBXL2-KO transient populations could be observed and no major protein degradation was observed in an FBXL2-AID/HA line upon treatment with auxin. However, no exflagellation defect could be detected upon auxin treatment preventing us from further functional analysis of this protein (Fig. 2k and Fig. S2d). We were, however, able to obtain a clonal FBXO1-AID/HA line (Fig. S2c) as well as a clonal line, FBXO1-GD (for Gene Disruption), in which 207 *fbxo1* bases were deleted, disrupting the coding sequence at the 353rd amino acid (Fig. S2h, j). Proteomic analysis indicated that seven FBXO1 tryptic peptides

were quantified in WT gametocytes, but only two of them were detected in the FBXO1-GD line, albeit with a significant sixfold decrease (Fig. S2j). Phenotypic analysis showed that FBXO1 is required for normal growth of asexual blood stages, microgametogenesis and ookinete development (See Fig. 2k for summary and following figures for more detailed analyses).

### FBXO1 shows dynamic apical localisation during schizogony and is important for apical complex assembly and merozoite segmentation

Given its expression in schizonts, we set out to further refine the localisation of FBXO1-HA during schizogony using U-ExM. We additionally stained cells for α and β Tubulin, the centrosome marker centrin and with an *N*-hydroxysuccinimide (NHS)-ester probe that reacts with primary amines for bulk proteome labelling[38,39]. During mitosis, FBXO1-HA is concentrated in a plate apically positioned above the centriolar plaque (Fig. 3a and Fig. S3a). Interestingly, an NHS-ester dense signal of unknown composition linked each FBXO1-HA-positive plate with a respective centrin signal. During the last round of mitosis, this position likely defines the future apical pole of the forming merozoite, where the apical complex and IMC are assembled. Later during merozoite segmentation, FBXO1-HA was enriched at the posterior pole of the growing subpellicular microtubules of the forming merozoite in a position that may correspond to the moving basal complex (Fig. 3b, c, Fig. S3b and Movie S1). Following nuclear segregation, the apical FBXO1-HA signal was fainter and dotted at the periphery of the merozoite (Fig. S3a, c). It also partly colocalised with MTIP (Fig. 3d, e), a previously described component of the glideosome machinery[40].

Following cloning of the FBXO1-GD line, we noticed a slight growth delay for asexual blood stages when passaging the line. We thus quantified parasitaemia of WT and FBXO1-GD asexual blood stages following intraperitoneal injection of 5,000 parasites. This confirmed that FBXO1-GD asexual blood stages proliferate slower than their WT counterpart (Fig. 3f). To further investigate the role of FBXO1 in asexual blood stages, we took advantage of the FBXO1-AID/HA line, speculating that the strong selective pressure following transfection of the *FBXO1-GD* targeting vector may have selected for adaptations to the disruption of *fbxo1*. Non-synchronised ring and trophozoite stages were grown in vitro in the absence or presence of auxin from the onset of the culture and phenotyped 24 h later (Fig. 3g). FBXO1-depleted schizonts displayed less nuclei that were not fully segregated from each other (Fig.3h, i and Fig. S3d). They also showed elongated and fragmented rhoptries, which were not linked to an apical polar ring (APR) as seen by NHS-ester staining of expanded schizonts. The subpellicular microtubules were also misshaped, making it difficult to count them, and did not radiate evenly from an APR in FBXO1-depleted schizonts. Electron microscopy analysis further confirmed the nuclear segregation defect as well as the dispersed rhoptries (Fig. 3j). It additionally revealed a segmentation defect with a single pellicle embedding multiple nuclei, a phenotype that was also observed with MTIP staining (Fig. 3k). Interestingly, the addition of auxin 8 h after the onset of schizont culture did not lead to these phenotypes (Fig. S3e), indicating that FBXO1 is required early during schizogony. Altogether these observations suggested that FBXO1 is important for nuclear segregation, assembly of the apical complex and merozoite segmentation during schizogony.

### SCF^FBXO1 expression during gametocytogenesis is required for gamete egress and microgamete centrosome partitioning

To further characterise requirements for FBXO1 during sexual development, we used U-ExM to refine FBXO1-HA, SKP1-HA, and FBXL2-HA localisation during microgametogenesis. FBXO1-HA, SKP1-HA and FBXL2-HA localised at the mitotic spindles and the axonemes but also showed a slight enrichment at the cell periphery (Fig. 4a, b and Fig. S4a, b), signals that were not observed in the WT control line.

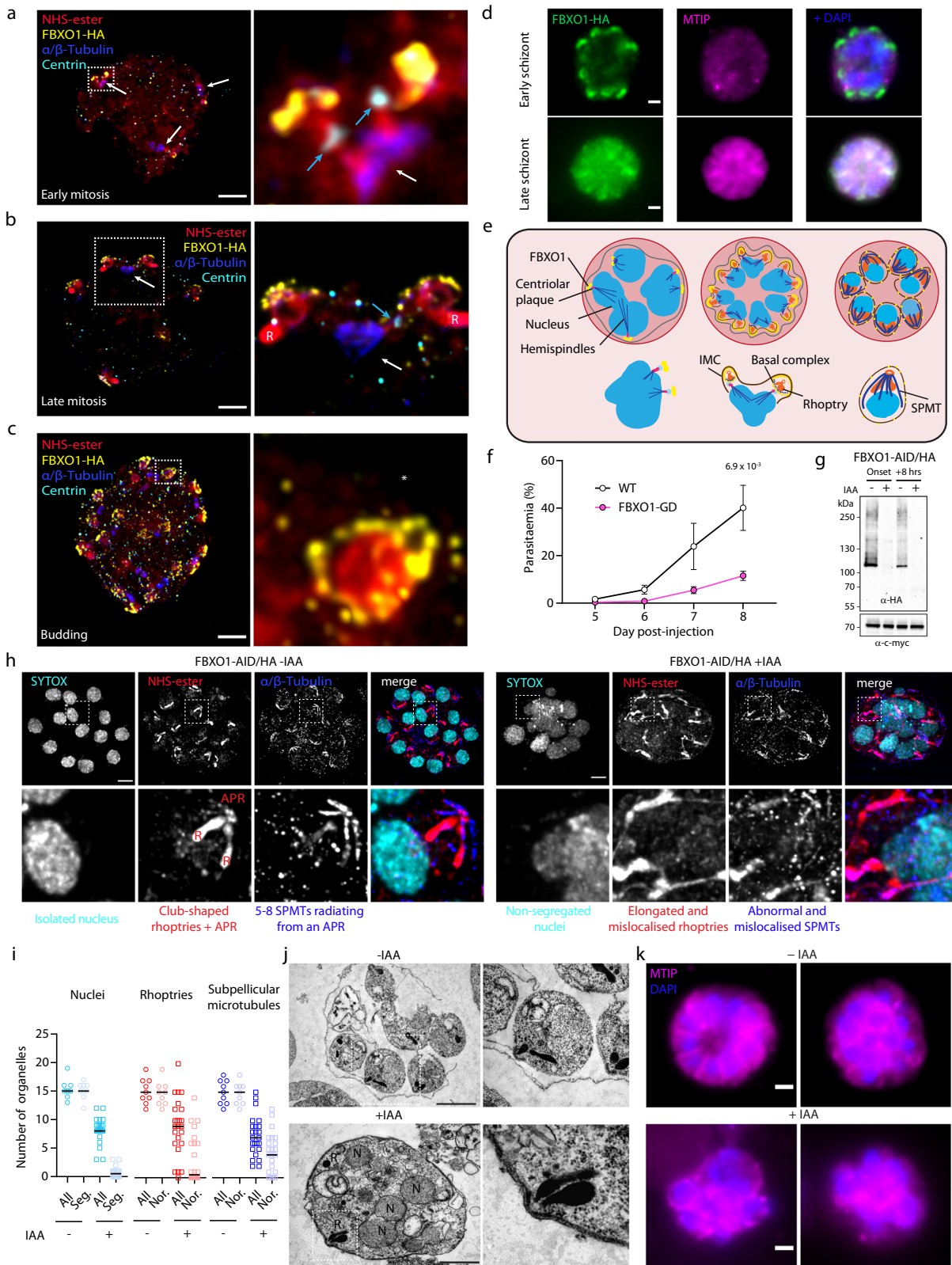

We then set out to define the cellular requirement for FBXO1 during sexual development. Disruption of *fbxo1* strongly impaired microgamete formation as determined by the percentage of microgametocytes leading to exflagellating microgametes (Fig. 4c). In contrast with the phenotype observed with FBXO1-GD microgametocytes, depletion of FBXO1-AID/HA in terminally differentiated gametocytes (Fig. 4d) did not significantly affect microgamete formation (Fig. 4e).

These results suggest that FBXO1 function in microgamete formation is required during the microgametocyte maturation prior to their activation by mosquito factors.

Microscopic observations of exflagellating FBXO1-GD parasites suggested that flagellar movement of the forming microgametes commenced while the parasite was still inside the host cell. To investigate membrane lysis of the host cell, we used a Ter-119 monoclonal

**Fig. 3 | FBXO1 displays dynamic localisation during late schizogony and links nuclear segregation with the formation of the apical complex. a–c** Confocal sections showing FBXO1-HA localisation during late schizogony depicting early mitosis (**a**), late mitosis (**b**) and merozoite segmentation (**c**) in expanded cells. The panels on the right show details of the boxed areas. White arrows indicate mitotic spindles, cyan arrows the Centrin-positive centriolar plaque, R the rhoptries. HA: yellow; α/β-Tubulin: dark blue; amine reactive groups/NHS-ester: red, centrin: cyan. Scale bars = 5 μm. **d** Localisation of FBXO1-HA (green) and MTIP (magenta) by widefield immunofluorescence in schizonts. DNA is stained with DAPI (blue). Scale bars = 1 μm. **e** Model depicting dynamic localisation of FBXO1-HA during late schizogony. **f** Reduced growth rate of FBXO1-GD asexual blood stages. Error bars show standard deviation from the mean; replicates from three independent infections; unpaired two-tailed *t*-test. **g** Western blot of FBXO1-AID/HA schizont lysates in the presence or absence of IAA added at different time points during the culture. Tir1-myc labelling serves as a loading control. **h** Effect of FBXO1-AID/HA depletion upon auxin addition on schizogony as observed by U-ExM. Lower panels show details of boxed merozoites. α/β-Tubulin: dark blue; amine reactive groups/NHS-ester: red, SYTOX (DNA): cyan. R rhoptries, APR apical polar ring. Scale bars = 5 μm. **i** Quantification of the nucleus, pair of rhoptries and sets of subpellicular microtubules (All) per schizont in the presence or absence of auxin. For each identified structure, we classified whether nuclei were segregated (Seg. – no overlap between DAPI positive surfaces), club-shaped rhoptries linked to an APR (Nor. for Normal), or 5–8 subpellicular microtubules emanating from an APR (Nor. for Normal) as seen in **h**; 33 cells from three biological replicates were analysed. **j** Electron micrographs showing nucleus (N), segmentation, and rhoptry (R) defects in FBXO1-AID/HA schizont upon treatment with IAA. Scale bars: 1 μm. **k** Localisation of MTIP (magenta) by widefield immunofluorescence in FBXO1-AID/HA schizonts in the presence or absence of IAA. DNA is stained with DAPI (blue). Scale bars: 1 μm. Source data are provided as a Source Data file.

antibody recognising a glycophorin-A-associated protein on the surface of the red blood cell. Seventy percent of FBXO1-GD gametocytes were still labelled twelve minutes post-activation, while nearly 100% of their wild type (WT) counterparts were extracellular at the same time point, indicating a delayed egress upon disruption of *fbxo1* (Fig. 4f). In addition, the flagella of FBXO1-GD microgametes appeared entangled within the remaining host erythrocyte. To investigate whether this was only dependent on the host cell membrane lysis, we compared mutant and WT gametocytes with U-ExM[41]. Co-staining of α/β-Tubulin and centrin highlighted delayed sequential segregation of the eight de novo-formed centrosomes in FBXO1-GD microgametocytes, while WT microgametocytes displayed segregated centrosomes (Fig. 4g, black arrows and Fig. S4c). This led to multiple axonemes remaining attached to each other even upon the onset of axoneme motility. Altogether, these results indicate that FBXO1 expression during gametocytogenesis is required for macro- and microgamete egress and centrosome partitioning during microgametogenesis.

### FBXO1-GD gametocytes show significantly reduced ubiquitination levels upon activation

We then set out to exploit the high synchronicity of developing gametocytes to identify whether the observed phenotypes of FBXO1-GD gametocytes were linked to defective ubiquitination by comparing their proteome and ubiquitome with those of WT parasites (Supplementary Data 3). To identify proximal ubiquitination events suggested by the rapid elevation of FBXO1 levels (Fig. 4d) concomitant with a rise in ubiquitin 48 signal (Fig. 1a), we focused our analysis on activated gametocytes 4 min following stimulation by XA. Apart from FBXO1 downregulation, IMC (ISP3 and IMC1g), basal complex (MORN1), glideosome (MTIP), glideosome-associated (GAP45) proteins were less detected in our proteomic survey (Fig. 4h). As these proteins are not known to be expressed in gametocytes, it suggests that they were less abundant in FBXO1-GD schizonts carried over during parasite purification. Interestingly, 181 lysine residues mapping on 151 proteins appeared significantly less ubiquitinated in FBXO1-GD gametocytes, and only a single lysine residue was found more abundant in FBXO1-GD gametocytes (Fig. 4i). GO term enrichment analysis indicated that multiple biological processes were differentially ubiquitinated upon *fbxo1* disruption including cell cycle and ubiquitin-dependent degradation (Fig. 4j and Supplementary Data 3). Given the pleiotropic nature of *fbxo1* and the large effect of its disruption on the gametocyte ubiquitome, we were not able at this stage to confidently predict ubiquitination events directly linked to FBXO1 requirements.

### FBXO1 is required to maintain the integrity of the pellicle and the subpellicular microtubule network in developing ookinetes

The peripheral enrichment of FBXO1-HA observed in gametocytes was much marked in developing and mature ookinetes, in which FBXO1-HA localised between the subpellicular microtubules and the plasma membrane (Fig. 5a). Consistent with this localisation, immunoprecipitation of FBXO1-HA in ookinete 12 h post-activation identified proteins associated with the pellicle (GAP45, GAP50, CTRP, SIP, ISC1, ISC3, GAPM1, GAPM2, Phil1, and MyoA) and alveolin proteins localising to the filamentous network below the IMC (IMC1b, c, d, e, f, h, I, j, k, l, and m) further confirming the peripheral localisation observed in ookinetes (Fig. 5b and Supplementary Data 2).

We then interrogated the requirement for FBXO1 during ookinete development. FBXO1-GD parasites did not form banana-shaped ookinetes and only retort stages were observed by Giemsa staining (Fig. 5c). Depletion of FBXO1-AID/HA one hour post-activation led to a similar phenotype, indicating that this phenotype is linked to FBXO1 requirement post-activation likely in the female lineage (Fig. 5d). We then examined FBXO1-GD ookinetes 20 h post-activation by U-ExM, which confirmed a late developmental arrest at the retort stage. Staining of expanded cells with α/β-Tubulin and NHS-ester indicated that the subpellicular microtubules were assembled but did not arrange in an evenly distributed network as observed in the WT ookinetes (Fig. 5e). Analysis of transmission electron microscopy sections of WT and mutant ookinetes further revealed abnormal IMC assembly in FBXO1-GD retort parasites where the IMC was either missing, incomplete or detached from the plasma membrane (Fig. 5f). Using CDPK1-HA as a pellicular marker in the developing ookinete[42], U-ExM revealed abnormal localisation of CDPK1-HA in FBXO1-GD parasites forming whorls, which were not observed in WT ookinetes (Fig. S5a, b). Altogether these results indicate a role of FBXO1 in maintaining the integrity of the pellicle and the subpellicular microtubule network during ookinete development.

### FBXO1 and CDPK1 share a molecular environment and show related cellular requirements during sexual development

While studying the cellular requirement for FBXO1 during gametocyte and ookinete development, we noted that the delayed egress and the production of retort ookinetes were partially reminiscent of the phenotypes previously described for parasites in which *cdpk1* expression was down-regulated[27]. Interestingly, CDPK1 was also slightly enriched in FBXO1-HA immunoprecipitates but not in those of SKP1-HA or RBX1-HA (Fig. 2c, f, g), suggesting a possible functional relationship between FBXO1 and CDPK1. As such a relationship could help to better define some of FBXO1 requirements, we first set out to confirm this observation by immunoprecipitating CDPK1-HA in gametocytes four minutes post-activation. This showed that the CUL1/RBX1/SKP1 complex and ISP1 were slightly enriched in CDPK1-HA immunoprecipitates (Fig. 6a). However, a large number of proteins were additionally enriched in CDPK1-HA immunoprecipitates only. This additional set was enriched in three distinct cellular components: the proton-transporting V-type ATPase complex, the small ribosomal subunit and the endocytic vesicle. As observed for FBXO1-HA, U-ExM confirmed that CDPK1-HA also localises to the ookinete periphery (Fig. 6b).

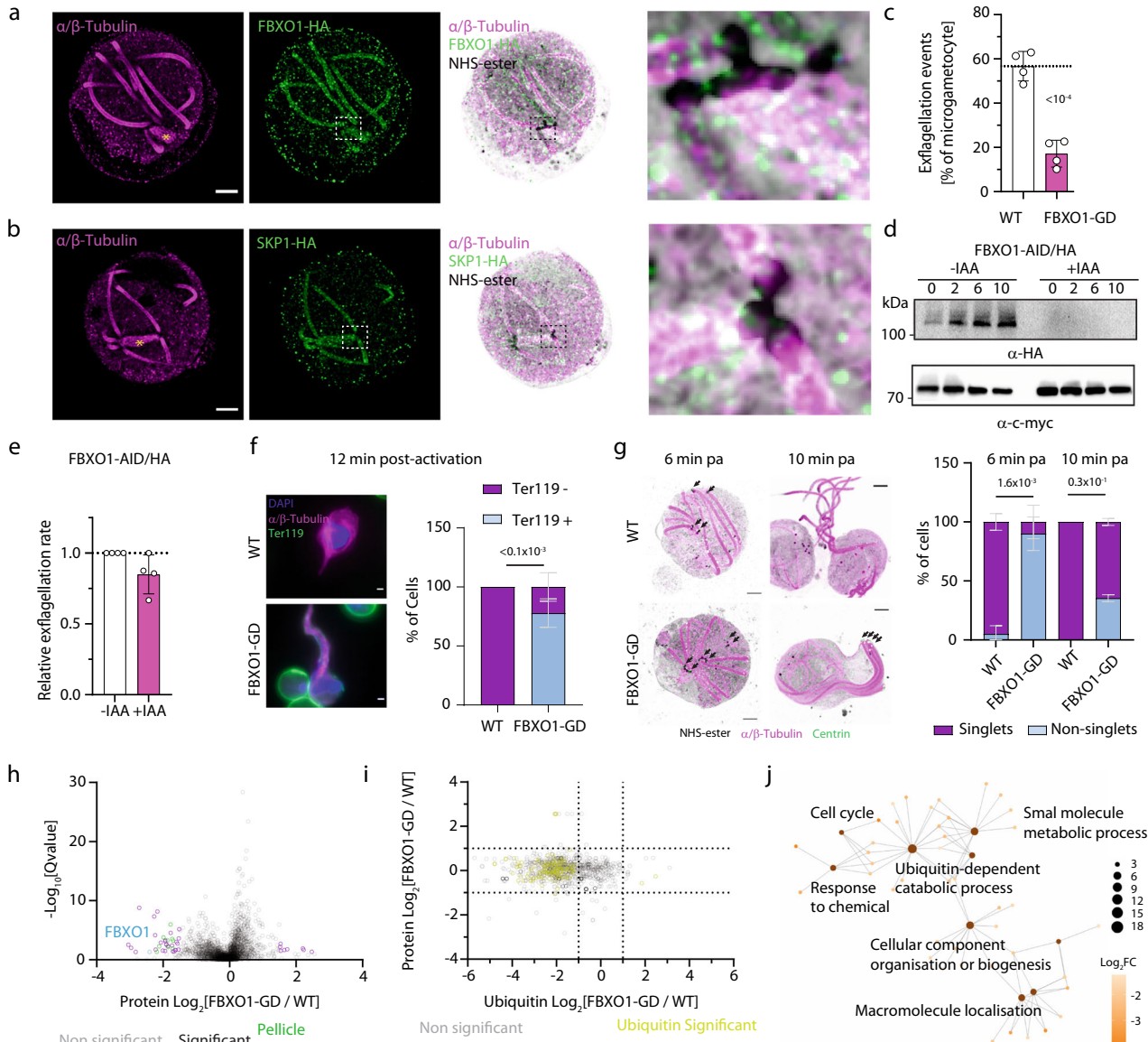

**Fig. 4 | FBXO1 is required for gamete egress, centrosome homoeostasis and protein ubiquitination during gametogenesis. a, b** Confocal section showing localisation of FBXO1-HA (**a**) and SKP1-HA (**b**) in activated and expanded *P. berghei* gametocytes. HA: green; α/β-Tubulin: magenta; amine reactive groups/NHS-ester: shades of grey. Asterisks indicate mitotic spindles. Insets show details of regions around microtubule organisation centres. Scale bars = 5 μm. **c** Effect of *fbxo1* disruption in gametocytes on exflagellation (error bars show standard deviation from the mean; replicates from four independent infections). **d** Western blot analysis of FBXO1-AID/HA gametocyte lysates over the course of gametogenesis in the presence or absence of IAA/auxin. Tir1-myc labelling serves as a loading control. **e** Effect of FBXO1-AID/HA depletion in gametocytes on exflagellation (error bars show standard deviation from the mean; replicates from four independent infections). **f** Gamete egress from host erythrocytes quantified by IFA based on the presence of the erythrocyte membrane marker Ter-119 (green), 12 min post-activation. Insets: widefield IFA of WT and FBXO1-GD gametocytes activated 12 min post-activation. Scale bars = 1 μm (56 cells analysed from three biological replicates,

error bars show standard deviation from the mean, one-way ANOVA). **g** Confocal section showing segregation of microgametocyte centrosomes by U-ExM. Quantification of the segregation defect is based on the identification of singlet or non-singlet centrin (green) and NHS-ester (shades of grey) positive centrosomes. α/β-Tubulin: magenta, scale bar = 5 μm (57 cells analysed from two biological replicates, error bars show standard deviation from the mean, one-way ANOVA). Black arrows indicate centrin-positive centrosomes; the grey arrow highlights the remnant membrane of the host erythrocyte. **h** Volcano plot showing differentially detected proteins in WT and FBXO1-GD gametocytes 4 min post-activation. Significantly regulated sites (*Q* value <0.05) with a fold change >2 are highlighted in blue (*n* = 3 biological replicates). **i** Plot indicating relative abundance of proteins (three biological replicates) and corresponding ubiquitinated peptides (technical duplicates from two biological replicates) in FBXO1-GD compared to WT gametocytes 4 min post-activation. **j** GO term enrichment analysis of differentially ubiquitinated proteins in FBXO1-GD compared to WT gametocytes 4 min post-activation. Source data are provided as a Source Data file.

Phenotyping of CDPK1-KO gametocytes[30] showed more marked defects compared to what was reported upon *cdpk1* downregulation[27] with a reduced exflagellation rate (Fig. 6c) and impaired egress (Fig. S6a). As for P_{ama1}CDPK1 parasites, we observed a complete block of ookinete development at the retort stage[27]. However, as opposed to FBXO1-GD ookinetes, CDPK1-KO ookinetes did not show defects in the

integrity of the IMC (Fig. S6b), highlighting non-overlapping requirements between CDPK1 and FBXO1. Altogether these results suggested that CDPK1 and FBXO1 have partially overlapping functions to control gamete egress, exflagellation and ookinete development. This raised the possibility of a possible interplay between CDPK1-dependent phosphorylation and FBXO1-dependent ubiquitination.

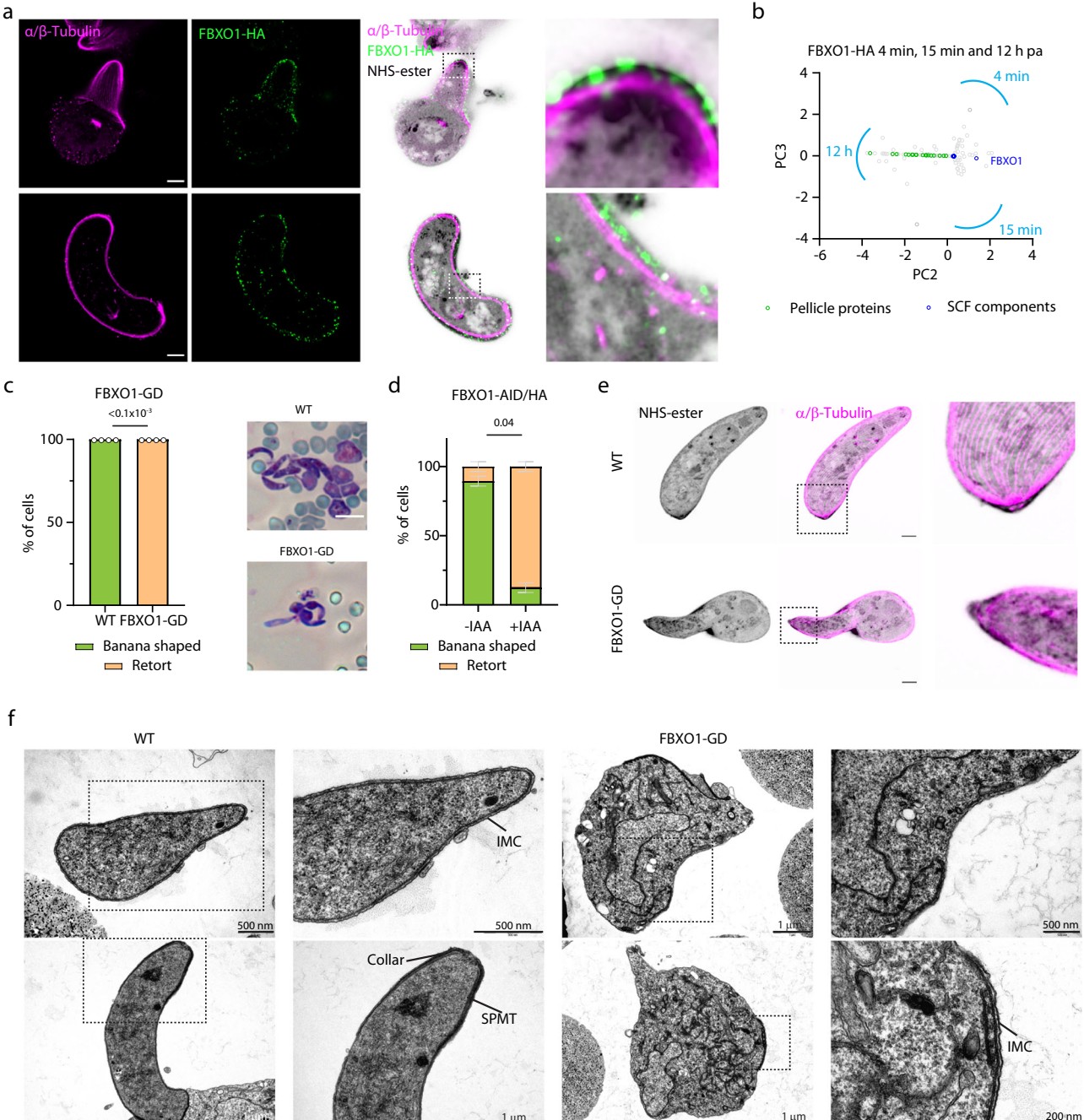

**Fig. 5 | FBXO1 is important for the assembly of the subpellicular microtubule network and IMC integrity during ookinete development. a** Confocal sections showing localisation of FBXO1-HA in the expanded retort (upper panels) and banana-shaped ookinetes (lower panels). HA: green; α/β-Tubulin: magenta; amine reactive groups/NHS-ester: shades of grey. Insets show details of the pellicle. Scale bars = 5 μm. **b** emPAI values as identified by mass spectrometry for proteins co-purifying with FBXO1-HA parasite, 4 min, 15 min and 12 h following activation with xanthurenic acid, and displayed in second and third principal components high-lighting enrichment of pellicular proteins during ookinete development (IP from one biological replicate). **c** Ookinete development of FBXO1-GD parasites as quantified by Giemsa staining (359 cells from four biological replicates, two-tailed student *t*-test). Images to the right show representative banana-shaped WT and retort FBXO1-GD ookinetes. **d** Effect of FBXO1-AID/HA depletion in terminally differentiated gametocytes on ookinete development quantified as in **c** (93 cells from two biological replicates, error bars show standard deviation from the mean, two-tailed student *t*-test). **e** Confocal sections showing expanded WT and FBXO1-GD ookinetes 20 h post-activation. α/β-Tubulin: magenta; amine reactive groups/NHS-ester: shades of grey. Scale bars = 5 μm. **f** Characterisation by transmission electron microscopy of WT and FBXO1-GD ookinetes 20 h post-activation. IMC inner membrane complex, SPMT subpellicular microtubules. Source data are provided as a Source Data file.

## CDPK1-K62 is ubiquitinated and is important for CDPK1 proteostasis in sexual stages

As ubiquitination provides a switching mechanism to turn on or off the activity of certain enzymes, such as kinases[43,44], we investigated whether FBXO1-dependent ubiquitination could regulate CDPK1 function during gametogenesis or ookinete development. Previous analyses have found that CDPK1 is ubiquitinated in asexual blood stages[13–15] and here we identified CDPK1-K62 to be ubiquitinated in gametocytes (Supplementary Data 1). Interestingly, this lysine is adjacent to the glycine triad, a highly flexible loop that points toward the catalytic cleft

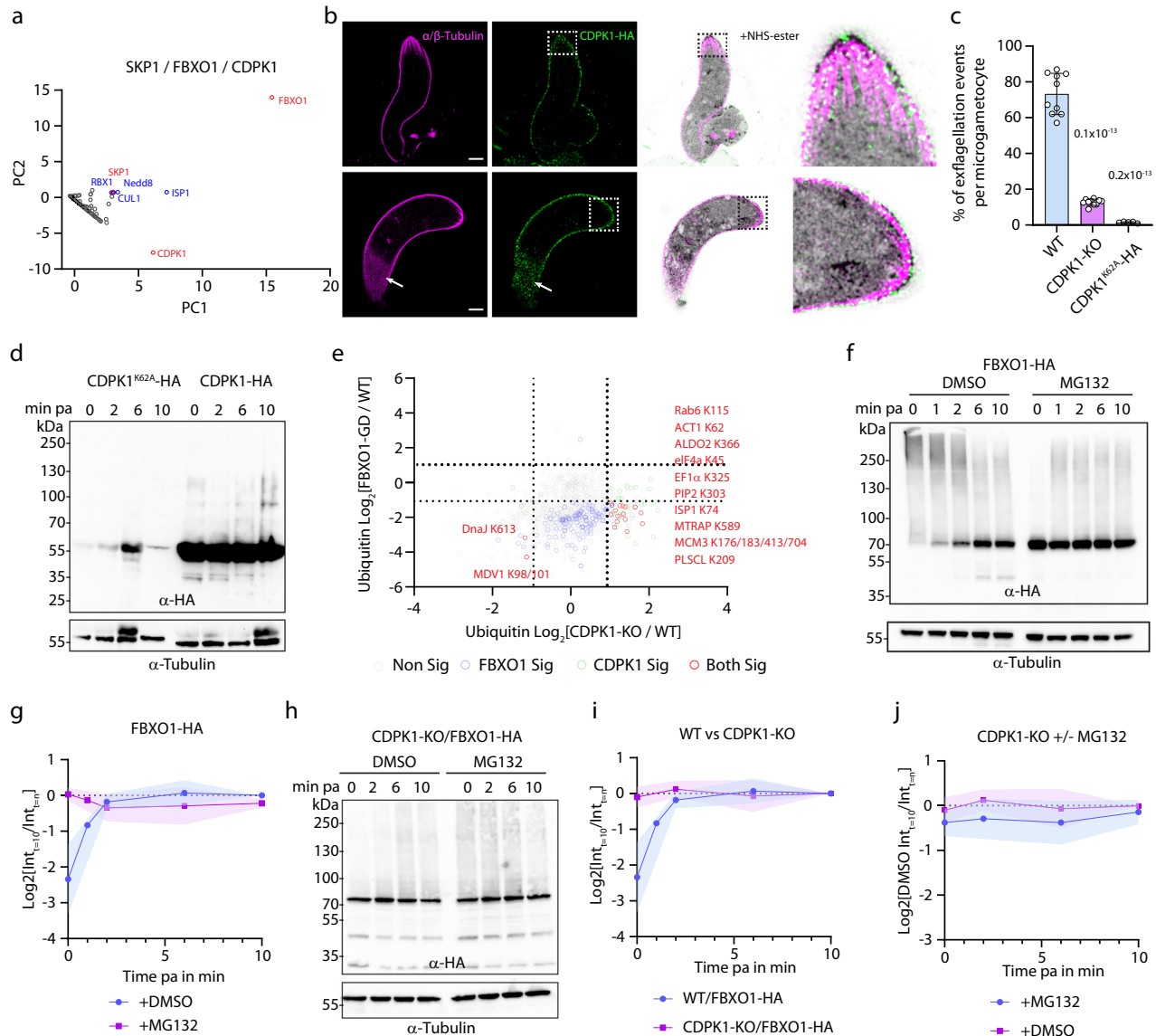

**Fig. 6 | Interplay between phosphorylation and ubiquitination during sexual development including regulation of FBXO1 proteostasis by CDPK1. a** emPAI values as identified by mass spectrometry for proteins co-purifying with SKP1-HA, FBXO1-HA and CDPK1-HA displayed in first and second principal components ($n = 2$ biological replicates). Red denotes immunoprecipitated proteins and blue possible components of the SCF complex. **b** Confocal sections showing localisation of CDPK1-HA in the expanded retort (upper panels) and banana-shaped ookinetes (lower panels). HA: green; α/β-Tubulin: magenta; amine reactive groups/NHS-ester: shades of grey. Insets show details of the pellicle. White arrows highlight the localisation of CDPK1-HA in the vicinity of the subpellicular microtubule network. Scale bars = 5 μm. **c** Effect of *cdpk1* deletion and CDPK1-K62 substitution to alanine on exflagellation (error bars show standard deviation from the mean; replicates from ten (WT and CDPK1-KO) and five (CDPK1$^{K62A}$-HA) independent infections; one-way ANOVA). **d** Western blot analysis of CDPK1-HA and CDPK1$^{K62A}$-HA gametocyte lysates. α-Tubulin serves as a loading control. **e** Plots showing the relative abundance of ubiquitinated peptides in FBXO1-GD gametocytes and CDPK1-KO

gametocytes (duplicates from two biological replicates). Sites significantly regulated in both mutants are highlighted in red. **f** Western blot analysis of FBXO1-HA gametocyte lysates in the absence or presence of 1 μM of the proteasome inhibitor MG132 added 30 min prior to activation. α-Tubulin serves as a loading control. **g** Quantification of FBXO1-HA abundance in the absence or presence of 1 μM MG132 added 30 min prior to activation from three independent biological replicates (coloured areas show standard deviation from the mean). **h** Western blot analysis of FBXO1-HA gametocyte lysates in WT and CDPK1-KO backgrounds. α-Tubulin serves as a loading control. **i** Quantification of FBXO1-HA abundance in WT and CDPK1-KO backgrounds from three independent biological replicates (coloured areas show standard deviation from the mean). **j** Quantification of FBXO1-HA abundance in the absence or presence of 1 μM of the proteasome inhibitor MG132 added 30 min prior to activation in the CDPK1-KO background from three independent biological replicates (coloured areas show standard deviation from the mean). Source data are provided as a Source Data file.

and functions to bind and position ATP. To investigate the functional relevance of lysine 62, the amino acid was substituted with an alanine residue in a CDPK1-HA marker-free line[30] (Fig. S6c). Two independent clones were readily obtained. These two clones showed strong exflagellation (Fig. 6c) and ookinete development defects (Fig. S6b), suggesting that K62 is important for the CDPK1 function. Western blot analyses showed strongly reduced levels of CDPK1$^{K62A}$-HA compared

with CDPK1-HA in developing gametocytes. This suggests that this substitution affects CDPK1 stability, likely explaining the observed phenotypes (Fig. 6d). As, CDPK1-K62 appeared slightly less ubiquitinated, but not in a significant manner, in FBXO1-GD gametocytes, we additionally checked whether CDPK1-HA levels could be dependent on FBXO1. However, no significant differences were observed by the western blot between WT and FBXO1-GD gametocytes (Fig. S6d).

Altogether, these results indicate that CDPK1-K62 is ubiquitinated in gametocytes and essential for CDPK1 function during sexual development. However, direct evidence for a regulation of CDPK1 by FBXO1-dependent ubiquitination could not be shown.

## CDPK1 negatively regulates FBXO1 function by downregulating its expression level in gametocytes

Phosphorylation can serve as a marker that triggers subsequent ubiquitination or that influences the activity of E3 ligases[44]. We thus wondered whether *cdpk1* deletion could be associated with patterns of differential ubiquitination as observed in FBXO1-GD gametocytes four minutes post-activation (Supplementary Data 4). No notable differences were detected between WT and CDPK1-KO proteomes apart from the absence of CDPK1 (Fig. S6e). In the Gly-Gly enriched fraction of CDPK1-KO gametocytes, 54 lysine residues mapping on 32 proteins were more ubiquitinated, while 10 lysine residues mapping on eight proteins were less ubiquitinated (Fig. S6f).

We then compared the changes among the 1037 ubiquitination sites that could be quantified in both CDPK1-KO and FBXO1-GD ubiquitomes. Strikingly, out of the 54 sites upregulated in CDPK1-KO gametocytes, 33% (18 sites) were found significantly less ubiquitinated in the FBXO1-GD line (Fig. 6e). Among those were two proteins localising to the pellicle (ISP1 and PIP2 - PBANKA_1013600), two proteins previously shown to be important for gamete egress (MTRAP - PBANKA_0512800 - and GEP - PBANKA_1115200), two proteins involved in DNA replication (DNA polδ -PBANKA_0501300- and MCM3 - PBANKA_1241800), a putative phospholipid scramblase (PLSCL - PBANKA_0506900) and actin I (PBANKA_1459300). This suggested that CDPK1 activity could be required to negatively regulate FBXO1-dependent ubiquitination of these residues that may represent important targets of FBXO1.

Interplays between phosphorylation and ubiquitination may involve phosphodegrons where phosphorylation serves as a mark for ubiquitination. However, these 18 inversely regulated sites appeared more ubiquitinated in the absence of CDPK1, suggesting that CDPK1-dependent activity is not a mark for their ubiquitination and subsequent degradation.

The regulation of biological processes by F-box proteins has been shown to depend on their loading onto their respective SCF complex or on their interaction with substrates[45–48]. We thus set out to investigate whether CDPK1 could be involved in regulating FBXO1 function by modifying FBXO1 loading on the SCF core or its interaction with possible substrates. To do so, we generated a transgenic line expressing triple HA-tagged FBXO1 in the CDPK1-KO background (Fig. S6g). Deletion of *cdpk1* did not significantly affect the recovery of SKP1, CUL1 and ISP1 in FBXO1-HA immunoprecipitates from activated gametocytes nor FBXO1-HA peripheral localisation (Fig. S6h, i). However, upon *cdpk1* deletion, we noticed that polyubiquitin was enriched and ubiquitin-60s ribosomal protein depleted in FBXO1-HA immunoprecipitates indicating that levels of these proteins in FBXO1-HA immunoprecipitates are dependent on CDPK1. However, these observations suggest that the loading of FBXO1-HA to the SCF is stable during gametogenesis and not dependent on CDPK1.

In human cells, F-box protein-dependent regulation has also been shown to depend on their dynamic expression[46,49–51]. We thus took advantage of the high synchronicity of gametogenesis upon induction by XA to assess FBXO1-HA levels during the first ten minutes of activation. FBXO1-HA showed increasing levels upon activation, with a peak reached between 2- and 6-min post-activation (Fig. 6f, g). Addition of MG132 prior to activation led to elevated levels of FBXO1-HA, which subsequently remained stable upon activation. This suggested active translation and degradation of FBXO1-HA prior to activation, possibly to prevent premature or over ubiquitination of substrate proteins in developing gametocytes. In contrast with WT gametocytes, FBXO1-HA detection levels remained high and stable prior to and upon

activation in CDPK1-KO cells, even in the absence of MG132 (Fig. 6h–j). We were not able to detect the ubiquitination of FBXO1-HA in our ubiquitinomics survey or by using various anti-ubiquitin antibodies on immunoprecipitated FBXO1-HA (Fig. S6j), and it remains unknown how FBXO1 expression levels are regulated in a CDPK1-dependent manner. Altogether, this suggests that CDPK1 activity is important to control FBXO1 levels in gametocytes, possibly by regulating its degradation.

## Discussion

Ubiquitination by CRL ligases regulates cell homoeostasis by modulating a wide range of cellular processes, frequently in association with other PTMs[47]. Recent studies have shown the relatively large extent of protein ubiquitination during the asexual replication of *Plasmodium* parasites in erythrocytes[14,15,19], but little is known about the enzymatic complexes mediating ubiquitination and their requirement during the *Plasmodium* lifecycle. Here we showed the conservation of an SKP1/CUL1/FBXO1 (SCF$^{FBXO1}$) complex in *Plasmodium*. This complex is required for multiple developmental stages in human and mosquito hosts. In human cells, SCF ligases recognise substrates via an adaptor module composed of SKP1 and one of ~68 F-box proteins[52]. Six additional cullin-RING complexes interact with distinct sets of adaptor modules, forming ~200 unique CRL complexes in total[53]. In *Plasmodium* gametocytes, we identified at least three possible adaptors interacting with SKP1: FBXO1, FBXL2 and Ank1, as also observed in *T. gondii*, a related apicomplexan parasite[54]. Interestingly, we observed overlapping localisation for SKP1, FBXO1 and FBXL2, and it is possible that disruption of *fbxo1* in erythrocytic stages may select for long-term complementation by FBXL2 or another adaptor protein. This could possibly explain the possibility to obtain an FBXO1-GD line while observing a strong defect in merozoite formation upon the rapid depletion of FBXO1-AID/HA in a single round of replication. Immunoprecipitation of SKP1-HA and RBX1-HA also detected other putative CRL components, including CUL2, further suggesting a degree of plasticity in *Plasmodium* CRLs, and further studies will be required to better understand the repertoire and roles of CRLs in *Plasmodium*.

Detailed cellular phenotyping of parasites lacking *fbxo1* or depleted of FBXO1 revealed a pleiotropic role at different stages of the *Plasmodium* lifecycle (Fig. 7). We found that *Plasmodium* FBXO1 is important for cell division by regulating nuclear segregation during schizogony and centrosome partitioning during microgametogenesis. In human cells, FBXO1, also named cyclin-F, has been shown to be a critical regulator of the cell cycle by controlling spindle formation and centrosome duplication[55]. If the latter process functionally links *Plasmodium* FBXO1 with other eukaryotic F-box proteins, an important substrate responsible for the requirement of FBXO1 in human centrosome segregation, the centriolar protein CP110[56], seems to be absent in *Plasmodium*. This suggests different wiring of FBXO1 in *Plasmodium* to regulate an apparently similar process, and more work will be required to better understand the molecular requirements of FBXO1 during *Plasmodium* cell division in asexual and sexual stages.

We provided evidence that FBXO1 is additionally required for biological processes that are specific to the intracellular parasitic lifestyle of *Plasmodium*. These include the gamete egress from the host erythrocyte and the integrity of two cellular structures important for zoite egress and invasion: the IMC and the apical complex of the merozoite and the ookinete. Interestingly, components of the IMC and the apical complex have been previously found to be enriched in two large ubiquitinomic surveys in the *P. falciparum* merozoite and in the *T. gondii* tachyzoite[15,57]. Here, using U-ExM, we showed that during schizogony, FBXO1 first localises at a site next to and defined by the centriolar plaque[58] where the IMC and the apical complex are formed. It then relocalises to the periphery of the merozoite, following a pattern reminiscent of the basal complex[59,60]. This intimate spatial and temporal association of FBXO1 with the centriolar plaque as well as the

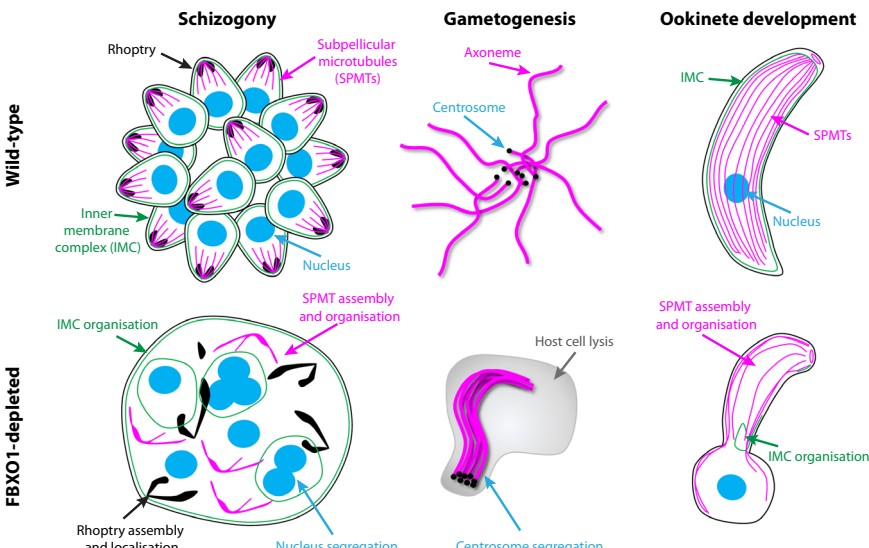

**Fig. 7 | Summary of the requirements for FBXO1 during schizogony, gametogenesis and ookinete development.** FBXO1-depleted schizonts show nuclear segregation defects, elongated and mislocalised rhoptries, an abnormal network of subpellicular microtubules (SPMT) and aberrant segmentation linked with a disorganised inner membrane complex (IMC). During gametogenesis, FBXO1 is important for host cell egress and segregation of the de novo-formed centrosomes. As observed in schizonts, ookinetes depleted of FBXO1 show an abnormal SPMT network and defects in the organisation of the IMC.

with the nascent apical complex, suggests an early role of the centriolar plaque in defining the polarity of the segmenting merozoites. The observed phenotypes suggest that FBXO1 is important in the following steps for proper assembly of the apical complex and the following merozoite segmentation (Fig. 7), as observed for the basal complex protein PfCINCH[59]. This localisation and requirements of FBXO1 during *Plasmodium* schizogony are highly reminiscent of those of TgFBXO1 during *T. gondii* endodyogeny, suggesting that despite the overall distinct modes of zoite formation, these two developmental stages show a similar functional requirement for FBXO1.

CRLs represent highly dynamic assemblies whose functions are regulated by several mechanisms, including CAND1-dependent sequestration, dynamic cullin neddylation and 'adaptor instability'[47]. By combining the FBXO1-AID/HA and the GD lines, we were able to identify at least three distinct windows of FBXO1 requirement: (i) towards the end of the last round of schizont nuclear division to form merozoites, (ii) before microgametocyte activation for microgamete development and egress and (iii) after macrogametocyte activation to form banana-shaped ookinetes. While we were unable to identify an orthologue of CAND1 that may have evolved beyond our limit of detection, Nedd8 was found enriched in RBX1-HA and SKP1-HA immunoprecipitates suggesting that neddylation is important to regulate SCF^FBXO1 function in *Plasmodium*, as previously suggested[61]. Interestingly, we noticed that FBXO1 levels are dynamically regulated, providing another level of regulation. Our results indicate that in non-activated gametocytes, low levels of FBXO1 are maintained in a CDPK1- and proteasome-dependent manner. The mechanisms behind these observations remain unknown but likely imply control of FBXO1 degradation. Interestingly, these results also imply that CDPK1 is active prior to gametocyte activation and the associated spike of calcium mobilisation[62,63]. Further work will be required to decipher the regulation of FBXO1 by CDPK1. It is important to note that while CDPK1 and FBXO1 functionally overlap, they do so only partially, making it more difficult to decipher their interplay. It is also likely that other PTMs are part of this functional relationship as, for example, acylation of both proteins was shown to be important for their function in *Toxoplasma* and *Plasmodium*[34,42,54,64]. CDPK1 is also ubiquitinated at a site that is important for its function but we did not find evidence that this modification depends on FBXO1. It nevertheless points to a tight interplay between ubiquitination, lipidation and phosphorylation to control development in *Plasmodium*.

Our comparative ubiquitinomic survey of FBXO1-GD gametocytes four minutes after gametocyte activation revealed a large number of ubiquitination events dependent on FBXO1. Whether this large number is directly or indirectly linked to FBXO1 activity remains unknown. By focusing on early-activated gametocytes, we may have missed or diluted out FBXO1-dependent ubiquitination events linked with its function in microgametogenesis as the related FBXO1 activity may be required prior to activation. We nevertheless detected FBXO1-dependent ubiquitination on two proteins important for egress, MTRAP and GEP, which may possibly regulate their function. We also revealed FBXO1-dependent ubiquitination on two IMC proteins, PIP2 and ISP1, suggesting that early ubiquitination of IMC components is important for ookinete development. While the role of PIP2 remains to be elucidated, ISP1 is one of the earliest known markers of ookinete polarity[26] and we also found it enriched in FBXO1 immunoprecipitates supporting the idea that ISP1 is a substrate of FBXO1 in *Plasmodium* sexual stages. A link between ISP1 and FBXO1 was also suggested to govern *T. gondii* tachyzoite formation[54]. However, ISP1 is not essential for the formation of *Toxoplasma* tachyzoites and *Plasmodium* ookinetes[25]. It is thus possible that ISP1 ubiquitination is not physiologically relevant for ookinete formation. However, compensation of *isp1* deletion by other ISP proteins may also mask ISP1 requirement for IMC biogenesis, and further investigations will be required to decipher the role of ISP1 ubiquitination.

Regulation by ubiquitination is well-known to be mediated through the ubiquitin-proteasome system, and it was proposed that increased ubiquitination of components of the IMC, glideosome and apical complex in the merozoite is linked to the destruction and recycling of these structures following invasion[15]. Here we show a link between ubiquitination and proteasome inhibition during gametogenesis. However, we noticed that FBXO1 activity also seems to be required prior to merozoite invasion and, more generally, early during schizogony, microgametogenesis and ookinete development. These requirements could be linked to early protein degradation to prevent premature and untimely regulation. It is also possible that FBXO1-dependent ubiquitination regulates the parasite development independently of the proteasome. Follow-up studies with time-resolved

ubiquitinomic surveys may help to identify ubiquitination events directly linked to FBXO1 activity and to study their functional relevance by reverse genetics. However, given the apparent large extent of ubiquitination, we anticipate that meaningful analyses may require to combine simultaneous analysis of multiple ubiquitination events with deep molecular phenotyping to link ubiquitination and cellular phenotypes.

# Methods

## Ethics statement
All animal experiments were conducted with the authorisation numbers GE102 and GE-58-19, according to the guidelines and regulations issued by the Swiss Federal Veterinary Office.

## Parasite maintenance and transfection
*P. berghei* ANKA strain[65]-derived clones 2.34[62] and 615[36], together with derived transgenic lines, were grown and maintained in CD1 outbred mice. Six- to 12-week-old mice were obtained from Charles River Laboratories, and females were used for all experiments. Mice were specifically pathogen-free (including *Mycoplasma pulmonis*) and subjected to regular pathogen monitoring by sentinel screening. They were housed in individually ventilated cages furnished with a cardboard mouse house and Nestlet, maintained at $21 \pm 2\,°C$ under a 12 h light/dark cycle, and given commercially prepared autoclaved dry rodent diet and water *ad libitum*. The parasitaemia of infected animals was determined by microscopy of methanol-fixed and Giemsa-stained thin blood smears.

For gametocyte and ookinete production, parasites were grown in mice that had been phenyl hydrazine-treated three days before infection. One day after infection, sulfadiazine (20 mg/L) was added to the drinking water to eliminate asexually replicating parasites. Microgametocyte exflagellation was measured three or four days after infection by adding 4 µl of blood from a superficial tail vein to 70 µl exflagellation medium (RPMI 1640 containing 25 mM HEPES, 4 mM sodium bicarbonate, 5% fetal calf serum (FCS), 100 µM xanthurenic acid, pH 7.4) in duplicates at least. An exflagellation event was defined as moving flagellated parasites forming clumps (exflagellation centres) with nearby red blood cells. To calculate the number of exflagellation centres per 100 microgametocytes, the percentage of red blood cells (RBCs) infected with microgametocytes was assessed on Giemsa-stained smears. For gametocyte purification, parasites were harvested in a suspended animation medium (SA; RPMI 1640 containing 25 mM HEPES, 5% FCS, 4 mM sodium bicarbonate, pH 7.20) and separated from uninfected erythrocytes on a Histodenz cushion made from 48% of a Histodenz stock (27.6% [w/v] Histodenz [Sigma/Alere Technologies] in 5.0 mM Tris HCl, 3.0 mM KCl, 0.3 mM EDTA, pH 7.20) and 52% SA, final pH 7.2. Gametocytes were harvested from the interface. To induce degradation of AID/HA-tagged proteins, 1 mM auxin dissolved in ethanol (0.2% final concentration) was added to purified gametocytes for one hour prior to activation by XA.

Ookinetes were produced in vitro by adding 1 volume of high gametocytaemia blood in 30 volumes of ookinete medium (RPMI 1640 containing 25 mM HEPES, 10% FCS, 100 µM xanthurenic acid, pH 7.5) and incubated at 19 °C for 18–24 h. Ookinete conversion efficiency was determined by Giemsa staining of ookinetes in duplicates at least. The conversion rate was determined as the percentage of banana-shaped or retort ookinetes.

Schizonts for transfection were purified from overnight in vitro cultures on a Histodenz cushion made from 55% of the Histodenz stock and 45% PBS. Parasites were harvested from the interface and collected by centrifugation at $500{\times}g$ for 3 min, resuspended in 25 µL Amaxa Basic Parasite Nucleofector solution (Lonza) and added to 10–20 µg DNA dissolved in 10 µl $H_2O$. Cells were electroporated using the FI-115 programme of the Amaxa Nucleofector 4D. Transfected parasites were resuspended in 200 µl fresh RBCs and injected intraperitoneally into mice. Parasite selection with 0.07 mg/mL pyrimethamine (Sigma) in the drinking water (pH ~4.5) was initiated one day after infection. A negative selection of parasites of FBXO1-GD parasites expressing yFCU was performed through the administration of 5 fluorocytosine (1 mg/mL, Sigma) via the drinking water[66]. Each mutant parasite was genotyped on a single genomic DNA preparation by PCR using three combinations of primers, specific for either the WT or the modified locus on both sides of the targeted region (experimental designs are shown in Supplemental Figures). For allelic replacements, sequences were confirmed by Sanger sequencing using the indicated primers. Parasite lines were cloned when indicated.

## Generation of DNA targeting constructs
The oligonucleotides used to generate and genotype the mutant parasite lines are in Table S1 and a summary of the background and generated parasite lines can be found in Table S2.

**Restriction/ligation cloning for the promoter swaps.** To generate the $P_{ama1}$ lines for genes of interest (*goi*) *cul1* and *rbx1*, the plasmid pOB116-ama1 was used[27]. The first -1000 bp of *cul1* and *rbx1* were amplified from genomic DNA using the primer pairs '$P_{ama1}goi$ HR1 forward' and '$P_{ama1}goi$ HR1 reverse'. The PCR products were Gibson assembled into the pOB116-ama1 plasmid digested with XhoI and EcoRV enzymes. The last ~750 bp of the goi 5′ UTR were amplified using primers '$P_{ama1}goi$ HR2 forward' and '$P_{ama1}goi$ HR2 reverse' and Gibson assembled into the modified pOB116-ama1 digested with HindIII and PstI enzymes. Plasmids were digested with the enzymes EcoRV and HindIII to be transfected in *P. berghei*. A schematic representation of the constructs and the recombined loci are shown in Fig. S2.

**PlasmoGEM vectors.** 3xHA, KO or AID/HA targeting vectors were generated using phage recombineering in *Escherichia coli* TSA strain with PlasmoGEM vectors (https://plasmogem.umu.se/pbgem/). For final targeting vectors not available in the PlasmoGEM repository, generation of knockout and tagging constructs were performed using sequential recombineering and gateway steps[67,68]. For each gene of interest (*goi*), the Zeocin-resistance/Phe-sensitivity cassette was introduced using oligonucleotides *goi* HA-F x *goi* HA-R and *goi* KO-F x *goi* KO-R for 3xHA, AID/HA tagging and KO targeting vectors, respectively. Insertion of the GW cassette following the gateway reaction was confirmed using primer pairs GW1 x *goi* QCR1 and GW2 x *goi* QCR2. The modified library inserts were then released from the plasmid backbone using NotI. The AID/HA targeting vectors were transfected into the 615 parasite line[36], while the KO/GD and triple HA targeting vectors were transfected into the 2.34 line unless otherwise specified. Schematic representations of the targeting constructs as well as WT and recombined loci, are shown in Fig. S2. Apart from the FBXO1-AID/HA line, the AID/HA lines were not cloned, and it is possible that more subtle phenotypes may be observed upon cloning despite the absence of noticeable regulation upon IAA treatment. In the case of the CUL1-HA line, as no signal could be detected above the background by western blot, we sequenced the full coding sequence, which revealed no mutation and confirmed that the HA coding sequence was in the reading frame. A second independent line was generated and gave similar results. As the protein is well detected by mass spectrometry, it remains unclear why no signals were observed by western blotting.

**Cas9 modifications.** The plasmid system previously used in *P. yoelii*, PYcm_PbU6 plasmid[69] which contains SpCas9, sgRNA and homology-directed donor template, was used for CRISPR/Cas9 modifications. gRNAs were designed using benchling (https://www.benchling.com/crispr). First, the PYcm_PbU6 plasmid was digested with BsmBI to insert the annealed gRNA, the guide RNA oligonucleotides (*cdpk1*$^{K62A}$ gRNA forward and reverse) were designed to generate overhangs for cloning into BsmBI. The annealed gRNA oligos, BsmBI restriction

digested plasmid and T4 DNA ligase (NEB) were incubated for 3 h at 16 °C for ligation. The ligated product was transformed into XL-10 Gold competent cells. Insert positive colonies were confirmed using 'cdpk1$^{K62A}$ gRNA forward and gRNA reverse sequencing universal primers'. PCR-positive colonies were sent for sequencing with the gRNA reverse sequencing universal primer.

A two-step PCR approach was used to generate the homology-directed repair (HDR) donor template. Homology region 1 (HR1, 558 bp) spanning 360 last bp of 5′UTR of *cdpk1* and the 197 first bp of *cdpk1* exon 1 was amplified from *P. berghei* genomic DNA, using primers 'cdpk1$^{K62A}$ HR1 forward' and 'cdpk1$^{K62A}$ HR1 reverse', which contained the modified bases to modify lysine 62 to alanine and the modified protospacer adjacent motif. Homology region 2 (HR2, 516 bp) spanning the region downstream to HR1 was amplified using primers 'cdpk1$^{K62A}$ HR2 forward' and 'cdpk1$^{K62A}$ HR2 reverse'. The PYcm_PbU6 plasmid containing the confirmed gRNA sequence was digested with HindIII and EcoRI to accommodate the homology-directed repair template. Finally, we used Gibson assembly to assemble HR1, HR2 and the linearised plasmid PYcm_PbU6 containing the gRNA to generate the final transfection vector. Oligonucleotides cdpk1$^{K62A}$ QCR1 to QCR6 were used to sequence the entire *cdpk1* gene in the clonal CDPK1$^{K62A}$-HA lines.

## Immunofluorescence assays

Cells were fixed with 4% formaldehyde and sedimented onto poly-D-lysine coated microscopy glass slides. Cells were briefly incubated with 0.2% TritonX-100 for 5 min and blocked with 3% bovine serum albumin for 1 h. Primary antibodies were added to the cells and incubated at 4 °C overnight in a humid chamber and was washed with phosphate buffer saline (PBS). Fluorescently labelled secondary antibodies were then added and incubated for 2–3 h at room temperature in the dark. Secondary antibodies were washed with PBS. Slides were then left to dry and cells were immediately mounted in Vectashield antifade mounting media with DAPI. Images were acquired on a widefield microscope Zeiss Axio Imager M2 using 100X objective. A list of antibodies and dyes, together with the used dilutions or concentrations, can be found in Table S2.

## Transmission electron microscopy

Sample preparation and imaging were performed as described in reference[30]. Ookinetes were fixed with 2.5% glutaraldehyde (Electron Microscopy Sciences) and 2.0% paraformaldehyde (Electron Microscopy Sciences) in 10 mM PBS pH 7.4 for 1 h at room temperature. Pelleted cells were embedded in 3% low melted agarose (Eurobio) in 10 mM PBS and dissected in small pieces in order for easier handling and to prevent loss of cells during subsequent processing steps. After extensive washing (five times 5 min) in 0.1 M sodium cacodylate buffer pH 7.4 (Sigma), samples were post-fixed with 1% osmium tetroxide (Electron Microscopy Sciences) reduced with 1.5% ferrocyanide (Sigma) in 0.1 M sodium cacodylate buffer pH 7.4 for 1 h at room temperature, followed by post-fixation with 1% osmium tetroxide alone (Electron Microscopy Sciences) in 0.1 M sodium cacodylate buffer pH 7.4 for one hour at room temperature. After washing with double distilled water (2 × 5 min), samples were en-block post-stained with 1% aqueous uranyl acetate (Electron Microscopy Sciences) for 1 h at room temperature. Samples were then washed with double distilled water for 5 min and dehydrated in graded ethanol series (2 × 50%, 1 × 70%, 1 × 90%, 1 × 95% and 2 × 100% for 3 min each wash). Samples were then infiltrated at room temperature with Durcupan resin (Sigma) mixed with 100% ethanol at 1:2, 1:1 and 2:1, each step for 30 min, followed by fresh, pure Durcupan resin for 2 × 30 min and transferred into fresh, pure Durcupan resin for 2 h. Finally, samples were embedded in fresh Durcupan resin-filled small thin-wall PCR tubes and polymerised and cured at 60 °C for 24 h. Ultrathin sections (60 nm) were cut with Leica Ultracut UCT microtome (Leica Microsystems) and diamond knife (DiATOME) and collected onto 2 mm single slot copper grids (Electron Microscopy Sciences) coated with 1% Pioloform plastic support film. Sections were then examined and TEM images were collected using Tecnai 20 TEM (FEI) electron microscope operating at 80 kV and equipped with a side-mounted MegaView III CCD camera (Olympus Soft-Imaging Systems) controlled by iTEM acquisition software (Olympus Soft-Imaging Systems).

## Expansion microscopy

U-ExM was performed as described in ref. [41]. Briefly, formaldehyde-fixed samples were attached on a 12 mm round poly-D-lysine (A3890401, Gibco) coated coverslips for 10 min. Coverslips were incubated for 5 h in 1.4% formaldehyde (FA)/ 2% acrylamide (AA) at 37 °C. Thereafter gelation was performed in ammonium persulfate (APS)/Temed (10% each)/Monomer solution (23% sodium acrylate; 10% AA; 0,1% BIS-AA in PBS) for 1 h at 37 °C. Gels were denatured for 1 h and 30 min at 95 °C. After denaturation, gels were incubated in distilled water overnight for complete expansion. The following day, gels were washed in PBS twice for 15 min to remove excess water. Gels were then incubated with primary antibodies at 37 °C for 3 h and washed three times for 10 min in PBS-Tween 0.1%. Incubation with secondary antibodies was performed for 3 h at 37 °C followed by three washes of 10 min each in PBS-Tween 0.1% (all antibody incubation steps were performed with 120–160 rpm shaking at 37 °C). Directly after antibody staining, gels were incubated in 1 ml of 594 NHS-ester (Merck: 08741) diluted at 10 µg/mL and 0.5 µM SYTOX™ Deep Red (Thermo Fisher Cat. No. S11381) in PBS for one hour and 30 min at room temperature on a shaker. The gels were then washed three times for 15 min with PBS-Tween 0.1% and expanded overnight in ultrapure water. Gel pieces of 1 cm × 1 cm were cut from the expanded gel and attached on 24 mm round poly-D-lysine (A3890401, Gibco) coated coverslips to prevent gel from sliding and to avoid drifting while imaging. The coverslip was then mounted on a metallic O-ring 35 mm imaging chamber (Okolab, RA-35-18 2000–06) and imaged. Images were acquired on a Leica TCS SP8 microscope with HC PL Apo 100x/1.40 oil immersion objective in lightning mode to generate deconvolved images. System-optimised z stacks were captured between frames using HyD as a detector. Images were processed with ImageJ and LAS X software. A list of used antibodies and dyes can be found in Table S2.

## Statistics and reproducibility

GraphPad Prism (9.5.1) was used for statistical analyses unless otherwise specified. Images shown in Figs. 1d, 5d and in Figs. 2b, e, h, i, 3c, d, j, k, 4b, 5a, f, 6b are representative of three and two independent infections, respectively. Blots shown in Figs. 1a, 2a, d, 6d, f, h and in Figs. 3g, 4d are representative of three and two independent infections, respectively. The number of replicates and statistical tests used are indicated in the figure legends.

## Immunoprecipitation

**Sample preparation.** Immunoprecipitations (IPs) of proteins were performed with purified gametocytes or ookinetes from two or three independent biological replicates, as previously described[70]. Samples were fixed for 10 min with 1% formaldehyde. Parasites were lysed in RIPA buffer (50 mM Tris HCl pH 8, 150 mM NaCl, 1% NP-40, 0.5% sodium deoxycholate, 0.1% SDS) and the supernatant was subjected to affinity purification with anti-HA antibody (Sigma) conjugated to magnetic beads. Beads were resuspended in 100 µl of 6 M urea in 50 mM ammonium bicarbonate (AB). Two microlitres of 50 mM dithioerythritol (DTE) were added and the reduction was carried out at 37 °C for 1 h. Alkylation was performed by adding 2 µl of 400 mM iodoacetamide for 1 h at room temperature in the dark. Urea was reduced to 1 M by the addition of 500 µl AB and overnight digestion was performed at 37 °C with 5 µl of freshly prepared 0.2 µg/µl trypsin (Promega) in AB. Supernatants were collected and completely dried under speed-vacuum.

Samples were then desalted with a C18 microspin column (Harvard Apparatus) according to the manufacturer's instructions, completely dried under speed-vacuum and stored at −20 °C.

**Liquid chromatography–electrospray ionisation tandem mass spectrometry (LC-ESI-MSMS).** Samples were diluted in 20 µl loading buffer (5% acetonitrile [CH₃CN], 0.1% formic acid [FA]) and 2 µl were injected into the column. LC-ESI-MS/MS was performed either on a Q-Exactive Plus Hybrid Quadrupole-Orbitrap Mass Spectrometer (Thermo Fisher Scientific) equipped with an Easy nLC 1000 liquid chromatography system (Thermo Fisher Scientific) or an Orbitrap Fusion Lumos Tribrid mass Spectrometer (Thermo Fisher Scientific) equipped with an Easy nLC 1200 liquid chromatography system (Thermo Fisher Scientific). Peptides were trapped on an Acclaim pepmap100, 3 µm C18, 75 µm × 20 mm nano trap-column (Thermo Fisher Scientific) and separated on a 75 µm × 250 mm (Q-Exactive) or 500 mm (Orbitrap Fusion Lumos), 2 µm C18, 100 Å Easy-Spray column (Thermo Fisher Scientific). The analytical separation used a gradient of $H_2O$/0.1% FA (solvent A) and $CH_3CN$/0.1 % FA (solvent B). The gradient was run as follows: 0 to 5 min 95% A and 5% B, then to 65% A and 35% B for 60 min, then to 10% A and 90% B for 10 min and finally for 15 min at 10% A and 90% B. Flow rate was 250 nL/min for a total run time of 90 min.

Data-dependant analysis (DDA) was performed on the Q-Exactive Plus with MS1 full scan at a resolution of 70,000 full width at half maximum (FWHM) followed by MS2 scans on up to 15 selected precursors. MS1 was performed with an AGC target of $3 \times 10^6$, a maximum injection time of 100 ms and a scan range from 400 to 2000 m/z. MS2 was performed at a resolution of 17,500 FWHM with an automatic gain control (AGC) target at $1 \times 10^5$ and a maximum injection time of 50 ms. The isolation window was set at 1.6 m/z and 27% normalised collision energy was used for higher-energy collisional dissociation (HCD). DDA was performed on the Orbitrap Fusion Lumos with MS1 full scan at a resolution of 120,000 FWHM followed by as many subsequent MS2 scans on selected precursors as possible within a 3 s maximum cycle time. MS1 was performed in the Orbitrap with an AGC target of $4 \times 10^5$, a maximum injection time of 50 ms and a scan range from 400 to 2000 m/z. MS2 was performed in the Ion Trap with a rapid scan rate, an AGC target of $1 \times 10^4$ and a maximum injection time of 35 ms. The isolation window was set at 1.2 m/z and 30% normalised collision energy was used for HCD.

**Database searches.** Peak lists (MGF file format) were generated from raw data using the MS Convert conversion tool from ProteoWizard. The peak list files were searched against the PlasmoDB *P.berghei* ANKA database (PlasmoDB.org[71], release 38, 5076 entries) combined with an in-house database of common contaminants using Mascot (Matrix Science, London, UK; version 2.5.1). Trypsin was selected as the enzyme, with one potential missed cleavage. Precursor ion tolerance was set to 10 ppm and fragment ion tolerance to 0.02 Da for Q-Exactive Plus data and 0.6 for Lumos data. Variable amino acid modifications were oxidised methionine and deamination (Asn and Gln) as well as phosphorylated serine, threonine and tyrosine. A fixed amino acid modification was carbamidomethyl cysteine. The Mascot search was validated using Scaffold 4.8.4 (Proteome Software) with a 1% of protein false discovery rate (FDR) and at least two unique peptides per protein with a 0.1% peptide FDR.

**Principal component analyses (PCA).** Proteins identified in the WT control from reference[72] were removed from the analysis. For PCA analysis, IPs of CRK5-HA performed under the same experimental conditions[72] were used as an additional control. The Exponentially Modified Protein abundance index (emPAI)[73] was calculated by Scaffold. Missing values were imputed with the minimum fixed value of 0.5. Replicates were averaged after log2 transformation and Principal

Component Analysis were computed and plotted using R. The protein–protein interaction network of enriched proteins was represented using Cytoscape 3.9.0.

**Proteomic analyses**

**Sample preparation for proteomic characterisation of gametocytes.** Cell lysis of three independent biological replicates was performed in 100 µl of 2% SDS, 25 mM NaCl, 50 mM Tris (pH 7.4), 2.5 mM EDTA and 20 mM TCEP supplemented with 1x Halt™ protease and phosphatase inhibitor. Samples were vortexed and then heated at 95 °C for 10 min with 400 rpm mixing with a thermomixer. DNA was sheared with four sonication pulses of 10 s each at 50% power. Samples were centrifuged for 30 min at 17,000×*g* and supernatants were collected. A Pierce protein assay was performed for each sample. Samples were then diluted with 200 µl of 50 mM Tris (pH 7.4) and incubated with 48 µl of iodoacetamide 0.5 M for 1 h at room temperature. Proteins were digested based on the FASP method using Amicon® Ultra-4, 30 kDa as centrifugal filter units (Millipore). Trypsin (Promega) was added at a 1:80 enzyme:protein ratio and digestion was performed overnight at room temperature. The resulting peptide samples were desalted with Pierce™ Peptide Deslting Spin Column (Thermo Fisher Scientific) according to the manufacturer's instruction; peptide concentration was determined using a colorimetric peptide assay (Thermo Fisher Scientific) and then completely dried under speed-vacuum.

**Enrichment of ubiquitinated peptides.** The PTM-Scan ubiquitin remnant motif (K-ε-GG) kit (Cell Signaling Technology, Kit #5562) was used and instructions described by the manufacturer were followed. For each ubiquitin remnant peptide enrichment immunoprecipitation (independent biological duplicates, each split in technical duplicates), -8–10 mg of total protein from purified gametocytes were used. Gametocytes were harvested from about 8–10 ml of *P. berghei*-infected blood in suspended animation and treated with 1 µm MG132 for 1 h at 37 °C. Ubiquitin remnant peptide enrichment was performed as described in ref. [74] with some modifications. Cell pellets were lysed in 5 mM iodoacetamide (IAA - I1149, Sigma-Aldrich) and 5% SDS/100 mM triethylammonium bicarbonate (TEAB - T7408, Sigma-Aldrich), and processed by an ultrasonic probe and heated at 90 C for 10 min. Samples were then processed again by ultrasonic probe five times 30 s on ice. Lysates were cleared by centrifugation at 16,000×*g* for 15 min. Protein concentration was measured with the Pierce 660 nm Protein Assay (Thermo), reduced with 10 mM tris(2-carboxyethyl)phosphine (TCEP – 77720, Thermo) at 56 °C for 15 min and alkylated by 10 mM IAA at RT for 30 min. Proteins were then precipitated by chloroform/methanol. One millilitre of 100 mM TEAB was added to the protein pellet and the mixture was left in an ultrasonic bath to disperse the pellet. One hundred micrograms of trypsin (Pierce) was added and incubated at 37 °C for 2 h. Another 100 µg of trypsin was added and incubated for a further 15 h. The protein digest was then heated at 70 °C for 10 min and dried in a SpeedVac. The PTMScan® IAP buffer was then added to the pellet, and the mix was further sonicated to dissolve the pellet. The sample was then centrifuged at 16,000×*g* at 4 °C to clear the lysate. To the cleared peptide digest, a vial of Cell Signaling Technology antibody-beads were added and incubated with rotation for 2 h at room temperature, followed by overnight incubation at 4 °C. The mixture was centrifuged at 2000×*g* for 30 s and ubiquitin-peptide enriched beads were washed twice with the IAP buffer and once with ice-cold HPLC water. Samples were finally incubated twice with 55 µl of 0.15% TFA for 10 min to elute ubiquitin-peptides. One hundred microliters of supernatant sample and 110 µL of IP (GlyGly enriched sample) were desalted with a C18 macrospin column and a C18 microspin column, respectively, (Harvard Apparatus, Holliston, MA, USA) according to manufacturer's instructions. A PierceTM Colorimetric Peptide assay (cat. No. 23275) was used to quantify the peptide amount after the desalting procedure. The total amount of

peptides was 24 and 68 µg in the GlyGly enriched and supernatant samples, respectively.

**ESI-LC-MSMS.** Duplicate samples were dissolved at 1 µg/µl with loading buffer (5% CH3CN, 0.1% FA). Biognosys iRT peptides were added to each sample and 2 µg of peptides were injected into the column. LC-ESI-MS/MS was performed on an Orbitrap Fusion Lumos Tribrid mass spectrometer (Thermo Fisher Scientific) equipped with an Easy nLC 1200 liquid chromatography system (Thermo Fisher Scientific). Peptides were trapped on an Acclaim pepmap100, C18, 3 µm, 75 µm × 20 mm nano trap-column (Thermo Fisher Scientific) and separated on a 75 µm × 500 mm, C18 ReproSil-Pur (Dr. Maisch GmBH), 1.9 µm, 100 Å, home-made column. The analytical separation was run for 135 min using a gradient of H2O/FA 99.9%/0.1% (solvent A) and CH3CN/H2O/FA 80.0%/19.9%/0.1% (solvent B). The gradient was run from 8% to 28% B in 110 min, then to 42% B in 25 min, then to 95%B in 5 min with a final stay of 20 min at 95% B. Flow rate was of 250 nL/min and total run time was of 160 min. Data-Independent Acquisition (DIA) was performed with MS1 full scan at a resolution of 60,000 (FWHM), followed by 30 DIA MS2 scans with fixed windows. MS1 was performed in the Orbitrap with an AGC target of $1 \times 10^6$, a maximum injection time of 50 ms and a scan range from 400 to 1240 m/z. DIA MS2 was performed in the Orbitrap using higher-energy collisional dissociation (HCD) at 30%. Isolation windows was set to 28 m/z with an AGC target of $1 \times 10^6$ and a maximum injection time of 54 ms.

**Data analysis.** DIA raw files for IP samples and Supernatant samples were loaded separately on Spectronaut v.15 (Biognosys) and analysed by directDIA using default settings (two.SNE files have been generated). Briefly, data were searched against *Plasmodium berghei* ANKA database (PlasmoDB.org, release 49, 5076 entries). Trypsin was selected as the enzyme, with two potential missed cleavage. Variable amino acid modifications were oxidised methionine and GlyGly (GG) derivatisation of lysine (+114 Da) (K). A fixed amino acid modification was carbamidomethyl cysteine. Both peptide precursor and protein FDR were controlled at 1% (Q value <0.01). Single Hit Proteins were excluded for Super Natant samples. For quantitation, the top 3 precursor area per peptides were used, "only protein group-specific" was selected as proteotypicity filter and normalisation was set to 'automatic'. A paired t-test was applied for differential abundance testing. The following parameters were used: Quantity at the MS2 level, quantity type = area, data filtering by Q value, and cross-run normalisation selected. Proteins and peptides were considered to have significantly changed in abundance with a Q value ≤0.05 and an absolute fold change FC ≥|1.5| (log$_2$FC ≥|0.58|).

**Enrichment analysis**
Gene Ontologies term enrichment analysis, as well as associated plots, were performed with ClusterProfiler[75,76] R package, using the EnrichGO function. Enrichment tests were calculated for GO terms, based on the hypergeometric distribution. P value cutoff was set to 0.05. The gofilter function was used prior to cnetplot drawing, to filter results at specific levels. Motif enrichment analysis was done using the motif-x algorithm proposed through the MoMo tool included in the MEME Suite[77]. FASTA files containing flanking sequences of interest were created using the seqRFLP R package. MoMo options were defined as follows: verbosity 1; width 15; eliminate-repeats 15; min-occurrences 5; score-threshold $1.0 \times 10^{-6}$. Results were exported in MEME Motif format and logo plots were created with motifStack R package.

**Softwares**
ImageJ 1.53, and LAS X (Leica version 3.5.7.23225) were used for image analysis. Excel (v1108) and GraphPad Prism 9 (9.5.1) were used for data and statistical analysis. Bio-Rad Image Lab (6.1) was used for western

blot analysis. Cytoscape (3.9.0) and R package were used for proteomic data analysis and representation. ClusterProfiler (3.8) was used for Gene Ontologies term enrichment analysis. Motif-x algorithm proposed through the MoMo tool included in the MEME Suite was used for motif enrichment analysis. Spectronaut v.15 (Biognosys) was used for DIA analysis. Mascot (Matrix Science, London, UK; version 2.5.1) was used for the peak list file searches against the PlasmoDB_P.berghei ANKA database (PlasmoDB.org, release 38). iTEM FEI (Olympus), ZEN 2.6 (Zeiss) and LAS X (Leica version 3.5.7.23225) were used for image acquisition.

**Reporting summary**
Further information on research design is available in the Nature Portfolio Reporting Summary linked to this article.

## Data availability
All data needed to evaluate the conclusions in the paper are present in the paper and/or the Supplementary Information. The mass spectrometry proteomics data generated in this study have been deposited to the ProteomeXchange Consortium via the PRIDE partner repository (http://proteomecentral.proteomexchange.org) with the dataset identifiers PXD035526 and PXD035557. Source data are provided with this paper.

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

## Acknowledgements

We thank the excellent service at the bioimaging (François Prodon, Nicolas Liaudet and Olivier Brun) and electron microscopy (Laura De Luca and Pilar Ruga Fahy) core facilities at the Faculty of Medicine (University of Geneva). We are also grateful to Nadia Walter and Alexandre Hainard for their excellent support at the proteomic core facility at the Faculty of Medicine (University of Geneva). We thank Ellen Bushell, Sophia Hernandez (Umeå universitet) and Jing Yuan (Xiamen University) for sharing the Cas9 vectors and for insightful advice regarding Cas9 editing. We thank Bohumil Maco (University of Geneva) for insightful advice on electron microscopy. We also thank Jyoti Choudhary and Lu Yu (Institute of Cancer Research) for their advice on ubiquitinomics. This work was supported by the Swiss National Science Foundation (31003A_179321 and 310030_208151) to M.B. M.B. is an INSERM and EMBO young investigator.

## Author contributions

R.R. and M.B. conceived the study; R.R. and N.K. performed the experimental work; D.S. and C.P. performed the proteomics analyses; R.R. and M.B. processed and analysed the results; M.B. sought funding for the work and wrote the initial manuscript. M.B., R.R., D.S. and C.P. reviewed and edited the manuscript.

## Competing interests

The authors declare no competing interests.
