## [Peer Review File · Nature Communications]

The Skp1-Cullin1-FBXO1 complex is a pleiotropic regulator required for the formation of gametes and motile forms in *Plasmodium berghei*REVIEWER COMMENTS

Reviewer #1 (Remarks to the Author):

This study presents the most comprehensive investigation to date into the role of SCF-Fbxo1 in Plasmodium parasite development. The authors use state-of-the-art proteomic and imaging approaches combined with extensive transgenics to provide a convincing picture of Fbxo1 involvement at multiple stages of parasite development. They very logically use the mouse malaria model, Plasmodium berghei, which lends itself more readily genetic manipulation than the human species, and also affords the ability to study the parasite across its vertebrate and invertebrate hosts. Moreover, given the high level of evolutionary conservation of the ubiquitin-proteasome pathway, it stands to reason that any findings in P. berghei will have direct implications on the more clinically relevant human Plasmodium species.

The authors convincingly show that SCF-Fbxo1 has pleiotropic functions across parasite development, playing essential roles in nuclear segregation and cell division during schizogony, parasite egress of the microgamete during gametogemesis, and organelle biogenesis essential to ookinete maturation. Furthermore, they present preliminary results suggestive of specific substrates the Fbxo1 E3 may be ubiquitinating in order to fulfil its cellular functions.

Overall, the authors have included an enormous amount of work, but at points the manuscript lacks justification as to why certain routes of exploration were pursued. This lack of detail results in a somewhat disjointed narrative that doesn't follow a logical sequence, making it more difficult to distil the important results- of which there are many!

For example, why did the authors choose to pursue the cdkp1 angle considering their ubiquitinomics screen highlighted likely Fbxo1 substrates (PIP2 ISP1 MTRAP GEP)? It seems they pursued multiple lines of research in parallel but the way the data are presented makes it seem like many were dropped before reaching a satisfiable concluding point. I'm not suggesting additional experiments, but perhaps a reorganisation following some of the suggestions detailed below could help do better justice to the very interesting and high quality data presented.

Finally, the manuscript is pitched to Plasmodium specialists and even more specifically to P. berghei ones. Less jargon and more explanation pertaining to some of the specialty markers, reagents and techniques used would help the narrative and make the whole study more accessible to the journal's wide readership.

Specific comments:

Figure 1- I'm confused by the panels in B. Why would the addition of MG132 serve to decrease the signal at 8'? Shouldn't it simply accumulate? Panel B is also missing the DMSO/MG132 labels

Figure 2- It would be helpful to include an explanation as to why the authors think they were unable to detect CUL1 by blot or IFA.

Figure 5A- could the slight enrichment of the SCF components simply be due to abundance and non-specific stickiness to HA beads? I realise the control is wild type lysate but could the authors compare the data against the IP of an unrelated HA-tagged protein at the same stages?

Figure 5G- have the authors considered whether the K62A point mutation might affect protein stability and thus explain the lower protein levels?

Figure 5 and accompanying text- I find the cdkp1 section/figure confusing as presented. I appreciate the amount of work that's gone into these experiments, but I'm struggling to distill what the main conclusions are and what the relevance to Fbxo1 is. I'd consider omitting this section altogether and keep the manuscript's focus on elucidating the role of Fbxo1 in parasite development rather than

identifying specific substrates.

Figure 6- I understand the reasoning behind making a KD version instead of using the KO line and fully agree. However, did that authors attempt to characterise the KO line during asexual development at all? Do they observe an effect on fitness in any way? A comparison of these two lines might provide further insight into the role of Fbxo1 during blood stage infection.

Minor: line

Line 243- 'human' should be corrected to 'humans'

Line 297- "we were not able to confidently predict proximal ubiquitination events linked to FBXO1 requirements." I don't understand what the authors are trying to say here. Could it be rephrased?

Line 393- replace 'foci' with 'focus'

Line 917- remove 'the'

Reviewer #2 (Remarks to the Author):

The manuscript titled "The Skp1-Cullin1-FBXO1 complex is a pleiotropic regulator required for the formation of gametes 1 and zoites in Plasmodium berghei" by Rashpa et al, aims to describe the location and function of proteins linked to the conserved SKP1/Culin1/FBXO1 complex.

While the work outlined is potentially exciting, unfortunately the data presented, falls well short of validating the conclusions of the authors.

There are a number of data presentation issues and inadequate explanation of the experiments and the rationale for doing them. Key controls are lacking in many instances.

With respect to the protein locations, the location of the proteins are interesting, however colocalization experiments with known markers of a range of cellular components are required to validate the proteins locations.

The overall structure of the paper could also be improved. For example it may be better to start with the protein characterisations in asexual stage parasites, then gametocytes, working through to gametes and ookinetes. In most instances there were little data discussed or displayed showing the phenotypes of the knockout or knockdown lines in asexual stage parasites and gametocytes. This is important to give context to the gamete and ookinete phenotypes reported.

Below I have outlined my detailed comments.

Detailed comments

Line 121 - In the text you state that the two proteasome inhibitors "target the catalytic site of the 26S proteasome." Given this would you expect an equivalent phenotype for the 2? There appears to be some differences in the phenotypes observed following treatment with the 2 inhibitors.

Figure 1 A – MG132 panel 15 min lane has not run well and should be repeated. It was not clear how many times these assays were done and if any quantification of the signal was done?

Figure 1B – There are no labels on the blots. Is the panel on the left DMSO and the panel on the right MG132?

Figure 1 C – Statistical information including, P values , number of cells etc are missing and should be included for all experiments.

Figure 1D - the images should have a separate letter. Ie be fig 1E. Also these images are not labelled. Are these images single slices or projection images? Round and stumpy look like single slices through the cells whereas retort and banana look like projections?

I am also concerned about assaying morphologies of cells following U-ExM preparation and imaging.

During this process there are a lot of factors that are a play that may effect the cellular structure. It would be better to present data from non-U-ExM experiments here. This will allow for large cell numbers to be investigated and will mitigate any issues associated with non-uniform expansion. Examples of the Untreated cell images should also be shown.

Line 130 – In the sentence starting “However, the subpellicular microtubule...” it is unclear what image/images this sentence refers to?

Figure 1E and F - In their current format figure 1 E and F add very little to the figure and manuscript. Individual proteins of interest should be labelled. Perhaps just a list of proteins would be better.

Table 1- The excel tables have very little explanation as to what is represented in each column and are also incompletely annotated. For example in Tab 1 of Table 1 excel sheet only a subset of the protein names are listed. A comprehensive legend should be added to the files. It was also not clear for the proteomics list of proteins what ordered they were present in in the table?

Line 141 – it is unclear why the 4- minute was chosen for the ubiquidomics?

Table 1 - Looking at the list of proteins identified in the Ubiquidomics. There appears to be a very large number of exported proteins that are identified? This isn't discussed or mentioned. How much overlap is there between different GO terms used here? And could this be simplified more to make it easier to display and discuss?

Figure 2 – No protein size labels on the figures. No controls for western blots. Unsure about why cul1-HA is included when it isn't expressed?

Figure S2 - With regards to the cell line validation shown in figure S2. I do not understand the validation PCRs. Looking at the diagram you have the goiQCR1 primer sitting in the middle of your 5' targeting sequence. If this is correct then the PCR with primer GW1 will amplify the plasmid sequence. In addition to this from my understanding of the diagram a PCR of QCR1 and QCR2 should yield a very large (2kb or more) PCR product if the plasmid is integrated and a small product if it isn't integrated or the cell line is a mixed population. In the gels displayed the FBXO-1-HA had a larger band and a smaller one?

The diagrams and gels for the AID-HA are also unclear for the same reasons discussed above. It also wasn't clear in the text if the cell lines being used were clonal lines or non clonal lines? The interpretation of KO and knockdown data from non clonal lines is problematic.

More clarification and validation of these cell lines are required.

There is no sizes associated with the DNA ladders.

Is the DNA ladder from the RBX1-AID/HA gel from a different gel? It appears that this is a different colour.

Figure S2C- The Western blots at the top of Panel C should be labelled, what do the 2 lanes represent? Full length blots should also be supplied and protein sizes. Also why are 2 of the bars in pink? I can't find reference to this.

Figure 2B – You need to show controls here, including non-transfected lines. You also need to show colocalization with a range of different markers to back up your claims in the text.

Figure 3 - The U-ExM images shown here show a significantly different localisation pattern to that shown in the initial images shown in 2B, where it was concluded to be at the mitotic spindle in activated male gametes.

Using conventional fluorescence techniques you can easily visualise microtubules and you would expect to see similar microtubule like staining in the images shown in fig 2B.

No Controls are supplied in either of the image sets. It would be good to see wildtype cells stained with the HA and secondary only controls to confirm that the signals being seen in the U-ExM are correct.

In addition to this conventional fluorescence images should be supplied using the same antibody sets and additional controls to confirm the cellular locations postulated in the text.

The zoom panels here should be separate and not in the corner as this is confusing in some panels and overlaps the main image in some images also.

Figure 3D – The location of FBXO1 at the cell periphery is interesting. It would be good to see some colocalization experiments with known IMC or parasite membrane markers to confirm the location of the protein.

Figure 3 I – Images of examples should be shown and the numbers of cells quantified should be reported.

Figure 4A – My understanding of this experiment is that the ter119 antibody will bind to the surface of RBCs so intact cells that have failed to egress but have undergone gametogenesis would be tubulin positive and ter119 positive, while wildtype will be tubulin positive and Ter119 negative as they have egressed from the RBC. Is that correct? If so the data reports that only 30% of the microgametes get trapped in the KO? Is this correct?

Figure 4B – it is extremely difficult to see the centrin labelling. For these experiments it would be good to use a marker of the parasite membranes and the RBC to determine what membranes are still intact. It is difficult to tell what membranes are still present using the NHS ester label.

More immunofluorescence microscopy using a range of markers against different parts of the mitotic machinery is required to conclude where the defect lies in the mitotic/cytokinesis process. While the U-ExM is excellent at viewing detailed structures, conventional microscopy and IFA approaches are better suited to confirming changes in morphology and loss of cellular structures.

Line 290 – “states a 6-fold reduction in FBXO1 detection” in the mass spec experiments. Given that the cell line is a KO shouldn't there be no FBXO1?

Figure 4D - The EM images need to be labelled so that the different structures are easily identifiable. Also these should not be relied upon as the sole piece of data for IMC formation, the EM images are single slices through a large cell. IFAs with known IMC proteins should be conducted and the morphologies observed and quantified.

Figure 6 - A large number of these images appear to be from schizonts that have already completed or have almost completed mitosis and or cytokinesis. The dense NHS stained structures at the apical end of the developing merozoites most likely represents the rhoptries, which has been observed in other U-ExM work suggesting that the parasites may be later in the development than thought.

The apical location of the FBXO1 is intriguing.

However it is not entirely clear from the images presented that it co-locates or is adjacent to centrin. It may be better to display 2 channels at a time, such as Centrin vs FBXO1 so these associations can be made clearer.

Co-location experiments with apical markers, centrin, IMC markers and plasma membrane markers could be done using conventional IFA and microscopy approaches to help validate the findings of the U-ExM. This would allow for larger numbers of cells to be analysed.

There also appears to be a bit of other binding from both the centrin and HA antibody in the U-ExM images. Is this non-specific or real? Controls of un-transfected lines may help with the HA antibody.

The differences between the FBXO1 KO and AID lines is interesting. Does the knockdown of FBXO1 lead to a growth defect? Growth assays should be supplied. Is it possible that the schizonts become delayed in development rather than arrested?

Line 433 – states...“we noticed that only SKP1-HA localised at the conoid complex of the merozoite. There is not enough evidence to state this.

Reviewer #3 (Remarks to the Author):

The manuscript by Rashpa et al. investigates the role of ubiquitination for life-cycle progression of the malaria-causing parasite and investigates the function of several molecular determinants. The experiments appear to be conducted according to highest standards and this study provides a very large amount of data of high quality. Because of this large amount of data, it is at times difficult to follow the argument. In addition, the data are sometimes presented in the “wrong” order, which makes it also difficult to follow.

Moreover, exciting biological insights, such as the interplay between phosphorylation and ubiquitination, are difficult to appreciate in the wealth of data. Thus, some data may be better provided as supplement, such as the crosses that suggested a maternal effect. This should be an easy fix and my comments below aim to help in this process.

L13/abstract, while the importance of the complex is clear, it is not solely exerting the control over the multiple different developmental processes. It would be more appropriate to claim that the complex is critical, key, etc. rather than claiming that the complex “controls” nuclear segregation, centrosome partitioning, gamete egress, apical complex, ookinete IMC etc, particularly in the abstract. For example, in line 58 and following, the description of the interplay between several regulatory mechanisms is balanced and the abstract would benefit from revision. The implicit claim that this complex alone controls all these processes is unrealistic and would require much more experiments.

L 35 and following, please provide a few more references, such as Kono et al., 2012.

L43 and title, the term “zoite” is Plasmodium-jargon and many readers will not understand, especially in context of the ookinete, which is a “zoit” despite the name. Please revise.

L76/77, for clarity, are there RING-finger E3s other than cullin-RING E3s. If yes, then the cullin-RING Es cannot be a family.

L79, when using the abbreviation APC/C, please also introduce the “cyclosome” in line 78.

For example, L94 or L100, the complex serves pleiotropic function and implicitly restricting this function to the processes that were analysed is probably an underestimation.

L117, Fig. 1A and B do not show a 10 but an 8-minute time point, please clarify. Also, A and B look very different and it would be worth harmonizing the labels, etc.

Fig. 1C Should the title not be “pre-treatment” as in the caption? Perhaps clearer would be “treatment pre-activation” and in Fig. 1D “treatment post-activation”.

Fig. 1D, the order of example images/categories and quantification thereof seems wrong, please reorder. Also, if 20 microtubule-containing cells per condition were analysed, showing the percentages of the different morphologies would be better. The presence of MTs is irrelevant, since these cells were chosen based on this.

L129, please guide the reader with arrow heads to the apical complex in Fig. 1D.

L145, IMC and glideosome are not in the list of enriched GO terms and this sentence would benefit from providing names of proteins that were more detected and which have a clear link to these structures. Also, please reorder GO terms and the volcano plot in Fig. 1E, as the GO enrichment is based on the data presented in the volcano plot.

Fig. 1G, the “G” seems hidden in my copy. Please check.

L164, 166, and elsewhere, please harmonize the mentioning of 3xHA-tags.

L166, again it is confusing, that data on FBXL2 and FBOX1 are shown in Fig. 2B although they are only identified as candidates later. Please revise. Also, the data on CUL1-HA is missing in Fig. 2B.

L174, why would only some proteins of the complex re-localize as shown in Fig. 2B? Please discuss.

Fig 2I, please explain in the caption what has been quantified. Cells per micro litre, field of view? Also, what is nd? Not done or non-detected?

L277, please comment and discuss the differences seen when comparing SKP1 localization in 2B and 3B?

L253, Reordering Fig. 2G and H would be more logical (first show that the protein can be depleted, then show the effect) and a reference to the data showing the downregulation upon auxin addition seems missing in the text.

Fig. 3I, how many cells were analysed in the two biological replicates? Please also provide statistics.

L264, this statement does not reflect what is shown in 4D. Here, it looks like only 30% are still Ter119 positive. Please check and provide statistics.

L269/Fig. 4B, while the shown image is very clear, a quantification of the enclosed vs. non-enclosed cells would be helpful. Perhaps the Ter119-staining in Fig. 4A can provide that but this is currently unclear. Also, the centrin signal is not visible and an inset with the centrin plus another stain would fix the problem. Thus, the quantification of singlet vs. non-singlet centrin signal is unclear. In addition, the text states that segregation of centrosomes is delayed, hence, should the singles not precede the non-singlet? Please revise.

L296, do you talk here about the gene or the protein? fbxo1 vs. FBOX1?

Fig. 5C, please explain what the “% of exflagellation canthers per microgametocytes” is?

Fig. 5D, as for Fig. 2I, please explain.

L341, Typo, “its” not “it”.

Fig. 6A, having the insets in a different orientation than the overview image makes it difficult to assess the signal in the bigger picture. Please revise. Similarly, in 6D. Please use the same frame and place a frame around the inset as well.

L406/Fig. 6F, images exemplifying “normal” and “non-segregated nuclei” and “elongated and fragmented rhopries” etc. are needed to assess the quantification of the phenotype in 6F. Please provide an image per category. If Fig. 6G should show these examples, please reorder the panels so the reader knows what has been quantified.

Fig. 6G, please provide where the insets have been cropped.

L411, please provide a reference where the corresponding imaging data can be found. Currently it is

unclear how uncoupled apical complexes and nuclei look like. Is this sentence based on the data shown in 6F? If so, it is unclear what “50% of observed apical complexes in FBXO1-depleted schizonts” mean. 50% comparing +IAA all vs mis.?

Reviewer #4 (Remarks to the Author):

General Comments

In the manuscript NCOMMS-22-30599 “The Skp1-Cullin1-FBXO1 complex is a pleiotropic regulator required for the formation of gametes and zoites in *Plasmodium berghei*”, the authors examine ubiquitination in SCF complexes. Overall I thought this was a good manuscript, well written, well presented and supported. One concern was Figure 1 and labelling errors and omission of labels. However, my main concern is that it appears only 2 biological replicates were used for the majority of the proteomic experiments. 2 biological replicates are not acceptable for the statistical analyses derived from the experiments. Thus the significance of the data is suspect.

Specific Comments

1. Figure 1 has mis-labels and discrepancies that make this section confusion and difficult to read. Line 117 – 119 describes figure 1A-B as anti-ubiquitin and anti-ubiquitin K-48. Figure 1A shows labels DMSO and 1 mM MG132 with s-UbR on the bottom, and 1B has labels anti-Ub K48 and anti UB-K48 for each. Please redo this figure with clear and consistent labeling. On line 881 there is mention of Figure 1G, but there is no G labeled on the Figure (I can assume which it is, but it should be labeled). And there is no label for the U-ExM images.
2. Figure S1 is a supplementary Western, the Ponceau control is not shown. The quality of the Western is questionable.
3. Line 877 regarding Figure 1E states 2 biological replicates were used. At minimum, 3 biological replicates is required to acquire meaningful statistics. I have major concerns about reporting these results with 2 biological replicates. Similar with line 948 regarding Figure 4E and line 959 Figure 5A where only two biological replicates are used, and Figure 5I/5K with only one biological replicate.
4. Over all the methods are very detailed. Please double check line 731. Matrix Science calculates the emPAI value. I am not aware that Scaffold does emPAI, I could be mistaken so please confirm this detail.
5. The biological replicate information is only disclosed in the Figure legends. The number of biological replicates should be described in detail in the Methods.

Reviewer #1 (Remarks to the Author)

This study presents the most comprehensive investigation to date into the role of SCF-Fbxo1 in Plasmodium parasite development. The authors use state-of-the-art proteomic and imaging approaches combined with extensive transgenics to provide a convincing picture of Fbxo1 involvement at multiple stages of parasite development. They very logically use the mouse malaria model, Plasmodium berghei, which lends itself more readily genetic manipulation than the human species, and also affords the ability to study the parasite across its vertebrate and invertebrate hosts. Moreover, given the high level of evolutionary conservation of the ubiquitin-proteasome pathway, it stands to reason that any findings in P. berghei will have direct implications on the more clinically relevant human Plasmodium species.

The authors convincingly show that SCF-Fbxo1 has pleiotropic functions across parasite development, playing essential roles in nuclear segregation and cell division during schizogony, parasite egress of the microgamete during gametogenesis, and organelle biogenesis essential to ookinete maturation. Furthermore, they present preliminary results suggestive of specific substrates the Fbxo1 E3 may be ubiquitinating in order to fulfil its cellular functions.

Overall, the authors have included an enormous amount of work, but at points the manuscript lacks justification as to why certain routes of exploration were pursued. This lack of detail results in a somewhat disjointed narrative that doesn't follow a logical sequence, making it more difficult to distil the important results- of which there are many!

We thank reviewer 1 for these positive and constructive comments. We agree that the narrative was disjointed and have completely reshaped the format of the manuscript and simplified the storyline.

For example, why did the authors choose to pursue the cdpk1 angle considering their ubiquitinomics screen highlighted likely Fbxo1 substrates (PIP2 ISP1 MTRAP GEP)? It seems they pursued multiple lines of research in parallel but the way the data are presented makes it seem like many were dropped before reaching a satisfiable concluding point. I'm not suggesting additional experiments, but perhaps a reorganisation following some of the suggestions detailed below could help do better justice to the very interesting and high-quality data presented.

We agree with reviewer 1. We are now explaining why we have decided to focus more on CDPK1 rather than putative substrates and how this could help to identify genuine substrates of SCF^{Fbxo1}.

Finally, the manuscript is pitched to Plasmodium specialists and even more specifically to P. berghei ones. Less jargon and more explanation pertaining to some of the specialty markers, reagents and techniques used would help the narrative and make the whole study more accessible to the journal's wide readership.

We again thank reviewer 1 for this comment. We have tried to make the text clearer for less specialist readers. We have changed the title accordingly.

Specific comments:

Figure 1- I'm confused by the panels in B. Why would the addition of MG132 serve to decrease the signal at 8'? Shouldn't it simply accumulate? Panel B is also missing the DMSO/MG132 labels

We apologize for the confusion. We have reannotated this panel (now Figure 1A). We hope it is clearer to reviewer 1 that, as expected, the ubiquitin signal increases over time in presence of MG132.

Figure 2- It would be helpful to include an explanation as to why the authors think they were unable to detect CUL1 by blot or IFA.

We thank reviewer 1 for this suggestion. We have first suspected a frameshift or a mutation leading to the absence of the HA tag expression. However, sequencing of the modified locus did not reveal any mutation that could explain the absence of HA expression. A second independent transgenic line showed the same absence of HA detection. As the protein is not detected by both IFA and western blotting, we suspect that the absence of signal is mainly linked to the low expression level. However, this explanation is not entirely satisfactory as the protein is well detected by mass spectrometry in our immunoprecipitates of FBXO1 and SKP1. We have now indicated this information in the method section of the manuscript. The text now reads:

“In the case of the CUL1-HA line, as no signal could be detected above the background by western blot, we sequenced the full coding sequence, which revealed no mutation and confirmed that the HA coding sequence was in frame. A second independent line was generated and gave similar results. As the protein is well detected by mass-spectrometry, it remains unclear why no signal is observed by western blotting”.

Figure 5A- could the slight enrichment of the SCF components simply be due to abundance and non-specific stickiness to HA beads? I realise the control is wild type lysate but could the authors compare the data against the IP of an unrelated HA-tagged protein at the same stages?

This analysis is shown in the new panels of Figure 2C and 2E that show enrichment of the SCF components compared with CRK5-HA immunoprecipitates (from Balestra et al, 2020). We are now making this information clearer in the method section. The text now reads:

“Proteins identified in the wild type control from reference [68] were removed from the analysis. For PCA analysis, IPs of CRK5-HA performed under the same experimental conditions [68] were used as an additional control.”

For information, we show below the number of peptides of the SCF components detected in immunoprecipitates (all duplicates) of different HA-tagged proteins under the same experimental conditions in gametocytes. This indicates that the enrichment of the SCF components in the IPs of CDPK1-HA, RBX1-HA, FBXL2-HA, FBXO1-HA and SKP1-HA over CRK5-HA IPs is also true when compared with 16 other HA-tagged proteins. In the manuscript, we have decided to show enrichment only over CRK5-HA as projection of 21 dimensions is less easy to visualise in 2 dimensions.

Protein	Kuehnel et al, in prep			Balestra et al 2021		Fang et al, 2017			Balestra et al, 2020		Brusini et al, 2022						This study					
	WT	GCα-HA	UGO-HA	PKG-HA	ICM1-HA	CDPK4-HA	SOC1-HA	SOC3-HA	SOC2-HA	CRK5-HA	ARK2-HA	Nek1-HA	Kin8X-HA	AKT1-HA	Spc24-HA	Nuf2-HA	AKT8-HA	CDPK1-HA	RBX1-HA	FBXL2-HA	FBXO1-HA	SKP1-HA
CUL1	0	0	0	2	0	5	0	0	0	3	0	0	0	0	0	5	0	4	253	9	4	89
NEDD8	0	0	0	0	0	0	0	0	0	0	0	0	0	0	0	1	0	0	32	1	1	7
RBX1	0	0	0	0	0	1	0	0	0	0	0	0	0	0	0	1	0	2	42	3	0	8
1358700	0	0	0	9	0	1	0	0	0	0	0	0	0	0	0	0	0	8	8	1	1	139
SKP1	0	0	0	0	0	0	0	0	0	0	0	0	0	0	0	0	0	0	6	0	4	17
FBXO1	0	0	0	0	0	0	0	0	0	0	0	0	0	0	0	0	0	17	0	0	131	69
FBXL2	0	0	0	0	0	2	0	0	3	5	5	2	2	0	0	3	0	0	145	15	4	62
ISP1	0	0	0	0	0	6	0	0	0	0	0	0	0	0	0	0	0	20	3	3	14	4
CDPK1	0	6	0	1	0	21	0	0	2	4	0	0	0	0	0	0	0	384	10	6	29	15

Figure 5G- have the authors considered whether the K62A point mutation might affect protein stability and thus explain the lower protein levels?

This is exactly what we hypothesise and thank reviewer 1 for noting we did not mention so. We have now modified the text to raise this hypothesis. The text now reads:

“This suggests that this substitution affects CDPK1 stability likely explaining the observed phenotypes”.

Figure 5 and accompanying text- I find the cdpk1 section/figure confusing as presented. I appreciate the amount of work that’s gone into these experiments, but I’m struggling to distill what the main conclusions are and what the relevance to Fbxo1 is. I’d consider omitting this section altogether and keep the manuscript’s focus on elucidating the role of Fbxo1 in parasite development rather than identifying specific substrates.

We understand that the previous version of the manuscript was likely not clear enough to highlight relevance of this data. We hope the new storyline of the manuscript makes the functional relationship between FBXO1 and CDPK1 clearer. We have also removed or moved to the supplementary materials less relevant information to focus on the most relevant findings.

Figure 6- I understand the reasoning behind making a KD version instead of using the KO line and fully agree. However, did that authors attempt to characterise the KO line during asexual development at all? Do they observe an effect on fitness in any way? A comparison of these two lines might provide further insight into the role of Fbxo1 during blood stage infection.

We thank reviewer 1 for this suggestion that was also proposed by reviewer 2. We have now quantified the evolution of the parasitaemia of the asexual blood stages in the FBXO1-GD line. This showed that FBXO1-GD parasites proliferate slower than their WT counterpart. These results have now been included in Figure 3F.

Minor:

Line 243- 'human' should be corrected to 'humans'

This was corrected.

Line 297- "we were not able to confidently predict proximal ubiquitination events linked to FBXO1 requirements." I don't understand what the authors are trying to say here. Could it be rephrased?

We have changed the sentence to "we were not able to confidently predict ubiquitination events directly linked to FBXO1 requirements".

Line 393- replace 'foci' with 'focus'

This was corrected.

Line 917- remove 'the'

We thank reviewer 1 for spotting this typo, we have removed "the".

Reviewer #2 (Remarks to the Author)

The manuscript titled “The Skp1-Cullin1-FBXO1 complex is a pleiotropic regulator required for the formation of gametes 1 and zoites in *Plasmodium berghei*” by Rashpa et al, aims to describe the location and function of proteins linked to the conserved SKP1/Culin1/FBXO1 complex.

While the work outlined is potentially exciting, unfortunately the data presented, falls well short of validating the conclusions of the authors.

There are a number of data presentation issues and inadequate explanation of the experiments and the rationale for doing them. Key controls are lacking in many instances.

With respect to the protein locations, the location of the proteins are interesting, however colocation experiments with known markers of a range of cellular components are required to validate the proteins locations.

The overall structure of the paper could also be improved. For example it may be better to start with the protein characterisations in asexual stage parasites, then gametocytes, working through to gametes and ookinetes. In most instances there were little data discussed or displayed showing the phenotypes of the knockout or knockdown lines in asexual stage parasites and gametocytes. This is important to give context to the gamete and ookinete phenotypes reported.

We thank reviewer 2 for these detailed and constructive comments as well as for the suggested controls. We now have taken these comments and suggestions into account and have included the missing controls or quantification. Please see our response to the detailed comments for more precisions. As suggested, by reviewers 1 and 3 we have also restructured significantly the manuscript.

Below I have outlined my detailed comments.

Detailed comments

Line 121 - In the text you state that the two proteasome inhibitors “target the catalytic site of the 26S proteasome.” Given this would you expect an equivalent phenotype for the 2? There appears to be some differences in the phenotypes observed following treatment with the 2 inhibitors.

We thank reviewer 2 for this comment. While both molecules have been described to target the 26S proteasome in other eukaryotes, they were also shown to display differential subunit specificity and others distinct non-proteasomal targets (see for example PMID 16778216). As the exact specificity of both molecules remains unknown in *Plasmodium*, we have decided to use both molecules at concentrations that were previously shown to inhibit proteasome degradation in *Plasmodium* gametocytes and/or during gametogenesis (see PMID 26118994 and 32568069 for 1 μ M MG132 and PMID 35130301 for 25 μ M bortezomib). We show that both molecules lead to an increase in ubiquitin signal in gametocytes and relatively similar phenotypes regarding male gamete and ookinete formation. The observed differences may thus be linked to different target specificities at the used concentrations or to different permeability. To avoid any confusion, we have removed the characterisation of the effects of 25 μ M bortezomib on exflagellation and ookinete development.

Figure 1 A – MG132 panel 15 min lane has not run well and should be repeated. It was not clear how many times these assays were done and if any quantification of the signal was done?

We agree with reviewer 2 that the lane did not run well. Despite our best efforts we have not managed to have nicely running blots showing all time points with the anti-ubiquitin antibody. We have however included blots of the lysates collected 10 min post-activation showing accumulation of the ubiquitin signal with α -Ub, α -Ub K48 and α -Ub K63 in Figure S1A. We are thus only keeping the blot performed with the anti-ubiquitin K48 in Figure 1A. For this blot, we also state that it is representative of three independent replicates in the figure legend.

Figure 1B – There are no labels on the blots. Is the panel on the left DMSO and the panel on the right MG132?

We apologize for the omission. We have reannotated Panel 1B that is now Panel 1A.

Figure 1 C – Statistical information including, P values, number of cells etc are missing and should be included for all experiments.

We have now included this information as suggested by reviewer 2 and 3. We have also included supplementary files with all raw values for each graph presented in this manuscript.

Figure 1D - the images should have a separate letter. I.e be fig 1E. Also these images are not labelled.

We have now labelled the images independently and moved them to Fig. S1C.

Are these images single slices or projection images? Round and stumpy look like single slices through the cells whereas retort and banana look like projections?

We thank reviewer 2 for spotting this missing information. We now indicate these are partial stack projections.

I am also concerned about assaying morphologies of cells following U-ExM preparation and imaging. During this process there are a lot of factors that are a play that may affect the cellular structure. It would be better to present data from non-U-ExM experiments here. This will allow for large cell numbers to be investigated and will mitigate any issues associated with non-uniform expansion. Examples of the Untreated cell images should also be shown.

We previously calculated the length to width ration before and after expansion and found no significant differences suggesting that the global morphology of cells is not affected by expansion (PMID 33705377). We now show quantification of the phenotype from non-expanded cells that show the same distribution in Fig. 1C-D. Using this approach, we did not include round forms that could also correspond to immature female gametocytes or residual trophozoites.

Line 130 – In the sentence starting “However, the subpellicular microtubule...” it is unclear what image/images this sentence refers to?

We have corrected this sentence to “Further investigation of the ookinete microtubule cytoskeleton by U-ExM indicated that treated parasites formed subpellicular microtubules (Fig. S1C).”

Figure 1E and F - In their current format figure 1 E and F add very little to the figure and manuscript. Individual proteins of interest should be labelled. Perhaps just a list of proteins would be better.

As suggested by reviewer 2, we have moved previous Figure 1E & 1F to Figure S1D & 1E. The list of proteins is available in Table S1.

Table 1- The excel tables have very little explanation as to what is represented in each column and are also incompletely annotated. For example in Tab 1 of Table 1 excel sheet only a subset of the protein names are listed. A comprehensive legend should be added to the files.

It was also not clear for the proteomics list of proteins what order they were present in in the table?

We thank reviewer 2 for this suggestion. We now label the columns as shown in the Figures and Supplementary Figures for all proteomic data. We are now listing the proteins and their ubiquitinated peptides based on the numerical value of the gene ID. We previously indicated the gene name whenever one was attributed in PlasmoDB. To avoid any confusion, we have removed this information that can easily be found in PlasmoDB.

Line 141 – it is unclear why the 4- minute was chosen for the ubiquitomics?

We thank reviewer 2 for spotting this oversight. We have added that “Samples were collected in non-activated gametocytes and four to six minutes post-activation, a time at which our western blot analysis suggested an increase in global ubiquitination levels (Fig. 1A).”

Table 1 - Looking at the list of proteins identified in the Ubiquitomics. There appears to be a very large number of exported proteins that are identified? This isn't discussed or mentioned

We thank reviewer 2 for this important suggestion. We are now mentioning this observation in the text: “Interestingly, a subset of exported proteins was also found to be ubiquitinated, raising the possibility that ubiquitination of parasite proteins also take place in the host cell.”

How much overlap is there between different GO terms used here? And could this be simplified more to make it easier to display and discuss?

GO term enrichment analysis was performed with ClusterProfiler R package, which allow filtering GO terms at selected levels to enable enrichment visualisation in details, or in general. The detailed GO term enrichment results are provided in the raw data files. For a clearer representation, the data was grouped at level 3 in the figures.

Figure 2 – No protein size labels on the figures. No controls for western blots. Unsure about why cul1-HA is included when it isn't expressed?

We apologise for including the wrong version of the figure in this submission. We have now included the protein size labels. As suggested by reviewer 1, we discuss in the method section our controls indicating that two independent clones showed no signals despite a correct coding sequence. The blot has now been moved to Figure S2B. The original image is also available in the file containing raw images, which shows that transfer of other proteins worked well for this specific blot. The related method section reads:

“In the case of the CUL1-HA line, as no signal could be detected above the background both by IFA or western blot, we sequenced the full coding sequence, which revealed no mutation and confirmed that the HA coding sequence was in frame. A second independent line was generated and gave similar results. As the protein is well detected by mass-spectrometry, it remains unknown why no signals are observed neither by IFA nor western blotting.”

Figure S2 - With regards to the cell line validation shown in figure S2. I do not understand the validation PCRs. Looking at the diagram you have the goiQCR1 primer sitting in the middle of your 5' targeting sequence. If this is correct then the PCR with primer GW1 will amplify the plasmid sequence. In addition to this from my understanding of the diagram a PCR of QCR1 and QCR2 should yield a very large (2kb or more) PCR product if the plasmid is integrated and a small product if it isn't integrated or the cell line is a mixed population. In the gels displayed the FBXO-1-HA had a larger band and a smaller one? The diagrams and gels for the AID-HA are also unclear for the same reasons discussed above.

The PlasmogEM team designed goiQCR1 and goiQCR2 so that their PCR product on non-modified genomic DNA is less than or around 1000 bp for the vectors we used in this study. In case of C-terminal tagging the goiQCR1 x goiQCR2 is in excess of 3,540 bp which is either not amplified under our experimental conditions or shows a ladder due to the 3'UTR repeats. In the case of a deletion, the sequence for goiQCR1 is deleted and no goiQCR1 x goiQCR2 is expected. We now indicate the expected product size for more clarity as well as the sizes of the cassettes encoding the tags and resistance marker in the schematics.

It also wasn't clear in the text if the cell lines being used were clonal lines or non clonal lines? The interpretation of KO and knockdown data from non clonal lines is problematic. More clarification and validation of these cell lines are required.

We are now better detailing which lines have been cloned in Table S7 and in the supplementary figures. AID/HA lines were not cloned for the initial screening. Following the initial screening, FBXO1-AID/HA was cloned. We now indicate this more clearly in the method section: Apart from the FBXO1-AID/HA line, the AID/HA lines were not cloned and it is possible that more subtle phenotypes may be observed upon cloning despite the absence of noticeable regulation upon IAA treatment. All knock-out, gene disruption, gene substitution, and promoter swap lines were cloned. CDPK1-HA and FBXO-HA lines were also cloned while other HA-tagged lines were not cloned. In addition, for all HA and AID/HA line (except CUL1-HA and CUL1-AID/HA), we also provide western blot and mass-spectrometry analyses showing correct expression of the tagged protein.

There is no sizes associated with the DNA ladders.

We are now indicating relevant marker sizes in the first panel of Figure S1A and Figure S5A. We indicate the same ladder has been used for all genotyping gels

Is the DNA ladder from the RBX1-AID/HA gel from a different gel? It appears that this is a different colour.

We agree with reviewer 2 that the colour is slightly different but the ladder and the RBX-AID/HA lanes are from the same gel. Below is a non-cropped version of the full gel.

We however think that reviewer 2 meant RBX1-HA and include below the full gel.

Figure S2C- The Western blots at the top of Panel C should be labelled, what do the 2 lanes represent? Full length blots should also be supplied and protein sizes.

Also why are 2 of the bars in pink? I can't find reference to this.

We thank reviewer 2 for spotting this oversight. We now indicate that the two bands correspond to gametocyte lysates in presence or absence of IAA. The full-length blots with protein sizes are now available in the supporting information.

We used pink for putative adaptor proteins as in Figure 2. As this was not clear and possibly confusing, all bars are now coloured in blue.

Figure 2B – You need to show controls here, including non-transfected lines. You also need to show colocalisation with a range of different markers to back up your claims in the text.

Western blot and immunoprecipitation experiments indicate that the proteins are expressed in gametocytes but do not allow to discriminate between male and female expression. In this figure, we mainly aim at confirming expression of SCF components in both sexes as U-ExM analysis is better suited to refine subcellular localisation – as indicated below we are now providing a WT control. We have thus toned down our claims regarding localisation at this stage of the manuscript.

Figure 3 - The U-ExM images shown here show a significantly different localisation pattern to that shown in the initial images shown in 2B, where it was concluded to be at the mitotic spindle in activated male gametes.

Using conventional fluorescence techniques you can easily visualise microtubules and you would expect to see similar microtubule like staining in the images shown in fig 2B.

No Controls are supplied in either of the image sets. It would be good to see wildtype cells stained with the HA and secondary only controls to confirm that the signals being seen in the U-ExM are correct.

We are now indicating that images shown in Figure 2 correspond to widefield microscopy, while Figure 4 (ex-Figure 3) are confocal sections following expansion microscopy. This makes it difficult to compare images for each figure given the gain of resolution brought by U-ExM. We thank reviewer 2 for indicating that we have not yet published a WT control for HA staining following cell expansion using the same acquisition parameters. We now show a WT control with α/β -Tubulin, HA and NHS-ester staining in Figure S4 indicating that the HA signal observed in Figure 4 is specific.

In addition to this conventional fluorescence images should be supplied using the same antibody sets and additional controls to confirm the cellular locations postulated in the text.

We think that the additional control shown in Figure S4 addresses this concern. Of note, we and others have previously published localisation of various other HA-tagged proteins that showed distinct localisation, including at the conoid, the IMC, nuclear membrane or kinetochore as determined by IFA, live microscopy and/or U-ExM. Please see PMID30315162, PMID36006241, PMID35900985, PMID33705377, PMID36109645 or PMID35077503 for recent examples.

The zoom panels here should be separate and not in the corner as this is confusing in some panels and overlaps the main image in some images also.

We have separated the zoom panels for all figures, as suggested by reviewer 2.

Figure 3D – The location of FXBO1 at the cell periphery is interesting. It would be good to see some colocalisation experiments with known IMC or parasite membrane markers to confirm the location of the protein.

We agree with reviewer 2 that this localisation is interesting. The NHS-ester staining indicates FBXO1-HA is less likely localising at the plasma membrane. The α/β -Tubulin staining also does not suggest any co-localisation with the subpellicular microtubules. IPs of FBXO1-HA recover proteins from the alveolin network, glideosome components and glideosome-associated proteins suggesting that FBXO1-HA is on both sides of the IMC. However, we do not have enough MTIP antibody for co-localisation experiments with U-ExM. So we prefer not to speculate to much about the exact localisation of FBXO1-HA. Of note, the exact localisation of multiple components of the glideosome including MTIP has still not been formally localised to the pellicular space between the IMC and the plasma membrane given the limits of most of the approaches used in the field.

Figure 3 I – Images of examples should be shown and the numbers of cells quantified should be reported.

We are now showing representative images of banana-shaped and retort ookinetes in Figure 5C.

Figure 4A – My understanding of this experiment is that the ter119 antibody will bind to the surface of RBCs so intact cells that have failed to egress but have undergone gametogenesis would be tubulin positive and ter119 positive, while wildtype will be tubulin positive and Ter119 negative as they have egressed from the RBC. Is that correct? If so the data reports that only 30% of the microgametes get trapped in the KO? Is this correct?

This is indeed the case. We thank reviewer 2 for spotting the discrepancy. The last author made a mistake when homogenising the colours across panels in the previous version. We have now corrected the figure, which is now Figure 4F.

Figure 4B – it is extremely difficult to see the centrin labelling. For these experiments it would be good to use a marker of the parasite membranes and the RBC to determine what membranes are still intact. It is difficult to tell what membranes are still present using the NHS ester label.

We have added black arrows to highlight the centrin-positive basal bodies. We agree with reviewer 2 that NHS-ester does not allow to specify which membrane is observed. As the Ter-119 staining in Figure 4F is already showing a defect in RBC membrane lysis, we have removed this conclusion from NHS-ester staining.

More immunofluorescence microscopy using a range of markers against different parts of the mitotic machinery is required to conclude where the defect lies in the mitotic/cytokinesis process.

We are not exactly sure which additional mitotic markers reviewer 2 is mentioning. For mitosis we include α/β -Tubulin that highlights the mitotic spindle, NHS-ester staining that stains the spindle pole (PMID35077503), and centrin that together with NHS-ester stains basal bodies that each links a spindle pole with an axoneme. To the best of our knowledge, we do not know published markers that could highlight cytokinesis in *P. berghei* gametogenesis. While it would be exciting to further refine the defect in FBXO1-GD microgametocytes, we here only describe a delay in centrosome segregation that explains the macro- and microscopic observations of bundled flagella, as assessed by exflagellation assay or α/β -Tubulin, centrin and NHS-ester staining.

While the U-ExM is excellent at viewing detailed structures, conventional microscopy and IFA approaches are better suited to confirming changes in morphology and loss of cellular structures.

We have recently published a detailed characterisation of gametogenesis or mitosis combining U-ExM, live microscopy and/or IFA and have rephenotyped multiple mutants that we and other groups previously characterised by IFA. This revealed multiple phenotypes that were missed by conventional IFA or gave much more structural details compared with conventional IFA (please see PMID35077503 and PMID36006241, for example). Similar observations were made by other groups to study mitosis in asexual blood stages (PMID 34835432 and PMID34535568). U-ExM also allowed us to identify a conoid in the *Plasmodium* ookinete that was previously not detected by IFA (PMID 33705377) and to further refine its substructure (PMID36109645). Therefore, we think that U-ExM is actually well suited to characterise changes in morphology and loss of cellular structures.

However, we agree that each approach has strengths and shortcomings. In addition to U-ExM, we have now added characterisation of FBXO1-depleted schizont by transmission electron microscopy and IFA with MTIP staining. These new experiments confirmed the nuclear segregation and rhoptry defects revealed by U-ExM but additionally showed a defect in segmentation that we now describe in the manuscript.

Line 290 – “states a 6-fold reduction in FBXO1 detection” in the mass spec experiments. Given that the cell line is a KO shouldn't there be no FBXO1?

We thank reviewer 2 for this important comment. The mutant line does not correspond to a full *fbxo1* deletion. It is a partial deletion that leads to disruption of the reading frame. To avoid any ambiguity, we renamed the line FBXO1-GD for Gene Disruption. We now describe the gene modification in more detail and precisely show in Figure S2J how this modification dramatically reduces FBXO1 expression:

“We were however able to obtain a clonal FBXO1-AID/HA line (Fig. S2C) as well as a clonal line, FBXO1-GD (for Gene Disruption), in which 207 *fbxo1* bases were deleted disrupting the coding sequence at the 353th amino acid (Fig. S2H and J). A proteomic analysis indicated that seven FBXO1 tryptic peptides were quantified in WT gametocytes but only two of them were detected in the FBXO1-GD line, albeit with a significant 6-fold decrease (Fig. S2J).”

Figure 4D - The EM images need to be labelled so that the different structures are easily identifiable. Also these should not be relied upon as the sole piece of data for IMC formation, the EM images are single slices through a large cell. IFAs with known IMC proteins should be conducted and the morphologies observed and quantified.

As suggested by reviewer 2, we have annotated the images and have also confirmed the IMC phenotype by EM and IFA using antibodies against MTIP from reference PMID16513191 as a pellicular marker for the schizonts (now Figure 3K). For the ookinetes, we took advantage of the FBXO1-GD/CDPK1-HA line to use CDPK1 as a pellicular/IMC marker (PMID28680058, 30315162 or 22817984 for example) to confirm the IMC defect observed by EM (Figure S5B).

Figure 6 - A large number of these images appear to be from schizonts that have already completed or have almost completed mitosis and or cytokinesis. The dense NHS stained structures at the apical end of the developing merozoites most likely represents the rhoptries, which has been observed in other U-ExM work suggesting that the parasites may be later in the development than thought.

We agree with reviewer 2 that the NHS-ester staining represents the rhoptries and we now clearly indicate this in the main text. For Figure 3A no such a staining is visible and the image likely shows early mitotic events. For Figure 3B and C, we however detect rhoptries together with mitotic spindles indicating that the final round of mitosis is not finished yet. We now better highlight the rhoptries and the mitotic spindles in Figure 3 to make this clearer.

The apical location of the FBXO1 is intriguing.

However it is not entirely clear from the images presented that it co-locates or is adjacent to centrin.

It may be better to display 2 channels at a time, such as Centrin vs FBXO1 so these associations can be made clearer.

Co-location experiments with apical markers, centrin, IMC markers and plasma membrane markers could be done using conventional IFA and microscopy approaches to help validate the findings of the U-ExM. This would allow for larger numbers of cells to be analysed.

As indicated by reviewer 2, NHS-ester stains the rhoptries that are part of the apical complex. We have simplified channels shown in Figure 3 to be clearer and highlight the centrin staining. It makes it clearer that FBXO1-HA does not co-localise with but is next to the centrin punctum. This was supported by the analysis a large number of forming merozoites (>30) and a representative gallery is available in Figure S3.

There also appears to be a bit of other binding from both the centrin and HA antibody in the U-ExM images. Is this non-specific or real? Controls of un-transfected lines may help with the HA antibody.

We agree with reviewer 2 that the centrin staining shows additional binding that we assume to be non-specific. Here we focus our analysis on centrin foci that are systematically adjacent to the mitotic spindles during schizogony (as previously described in PMID34535568 or PMID33705377) or to centrin foci colocalising with a strong NHS-ester signal at the base of axonemes in gametocytes (as described in PMID35077503). The same situation is observed with a low and scattered background observed with the HA antibody as shown in Figure S4. Although the background is

relatively low, co-staining with additional markers such as NHS-ester and α/β -Tubulin clearly highlights stronger and specific staining for all analysed proteins. As mentioned above, we and others have previously published localisation of various other HA-tagged proteins that showed distinct localisation, including the conoid, the IMC, nuclear membrane or kinetochore as determined by IFA, live microscopy and/or U-ExM. Please see PMID30315162, PMID36006241, PMID35900985, PMID33705377, PMID36109645 or PMID35077503 for recent examples. The apical and peripheral localisation described here is thus unlikely due to unspecific binding. We can of course not rule out that a minor population of FBXO1-HA that does not reach a signal above the background fluorescence is localised in others subcellular compartments.

The differences between the FBXO1 KO and AID lines is interesting. Does the knockdown of FBXO1 lead to a growth defect? Growth assays should be supplied. Is it possible that the schizonts become delayed in development rather than arrested?

We have now quantified the evolution of the parasitaemia of the asexual blood stages in the FBXO1-GD line. This shows that FBXO1-GD parasites proliferate slower than their WT counterpart indicating an important role of FBXO1 for the growth of asexual blood stages. These results have now been included in Figure 3F. For FBXO1-AID/HA schizonts treated with IAA, the rhoptries and SPMT phenotypes do not correspond to a normal intermediate developmental stage indicating that the observed phenotype is unlikely to correspond to a delayed growth. The new EM images and MTIP IFAs provided in Figure 3J/K also support this conclusion.

Line 433 – states...”we noticed that only SKP1-HA localised at the conoid complex of the merozoite. There is not enough evidence to state this.

We agree with reviewer 2 that this was a premature statement and have decided to remove this observation to better focus on FBXO1.

Reviewer #3 (Remarks to the Author)

The manuscript by Rashpa et al. investigates the role of ubiquitination for life-cycle progression of the malaria-causing parasite and investigates the function of several molecular determinants. The experiments appear to be conducted according to highest standards and this study provides a very large amount of data of high quality. Because of this large amount of data, it is at times difficult to follow the argument. In addition, the data are sometimes presented in the “wrong” order, which makes it also difficult to follow.

Moreover, exciting biological insights, such as the interplay between phosphorylation and ubiquitination, are difficult to appreciate in the wealth of data. Thus, some data may be better provided as supplement, such as the crosses that suggested a maternal effect. This should be an easy fix and my comments below aim to help in this process.

L13/abstract, while the importance of the complex is clear, it is not solely exerting the control over the multiple different developmental processes. It would be more appropriate to claim that the complex is critical, key, etc. rather than claiming that the complex “controls” nuclear segregation, centrosome partitioning, gamete egress, apical complex, ookinete IMC etc, particularly in the abstract. For example, in line 58 and following, the description of the interplay between several regulatory mechanisms is balanced and the abstract would benefit from revision. The implicit claim that this complex alone controls all these processes is unrealistic and would require much more experiments.

We thank reviewer 3 for these important suggestions. We have modified the abstract and line 58 accordingly:

Abstract: It is key to cell division for nuclear segregation during schizogony and centrosome partitioning during microgametogenesis. It is additionally required for parasite-specific processes including gamete egress from the host erythrocyte, as well as formation of the merozoite apical complex and the ookinete inner membrane complex (IMC), two structures essential for *Plasmodium* dissemination. Ubiquitinomic surveys reveal a large set of proteins ubiquitinated in a FBXO1-dependent manner including proteins important for egress and IMC organisation. We additionally demonstrate bidirectional interplay between ubiquitination and phosphorylation via calcium-dependent protein kinase 1. Altogether we show that *Plasmodium* SCF^{FBXO1} plays conserved roles in cell division and is also important for parasite-specific processes in the mammalian and mosquito hosts.

Line 58: Transcriptional regulation has been shown to play an important part in coordinating these multiple phases of parasite development [8, 9]. However, key aspects of cellular development also require post-translational modifications (PTMs) of proteins [10]. Multiple PTMs have been shown to be important for developmental transitions including phosphorylation, acetylation, methylation, and lipidation [10-12] but little is known about the requirement of ubiquitination in *Plasmodium*.

L 35 and following, please provide a few more references, such as Kono et al., 2012.

We thank reviewer 3 for pointing to this important reference that was indeed missing.

L43 and title, the term “zoite” is Plasmodium-jargon and many readers will not understand, especially in context of the ookinete, which is a “zoit” despite the name. Please revise.

We agree the term zoite is *Plasmodium* jargon and confusing when it comes to ookinetes. We have changed these two occurrences by “motile forms”.

L76/77, for clarity, are there RING-finger E3s other than cullin-RING E3s. If yes, then the cullin-RING Es cannot be a family.

We thank reviewer 3 for spotting this typo, we have modified the sentence accordingly: “Among the RING-finger ligases, the cullin-RING E3 ligases (CRLs) represent a diverse group that includes ligases such as the Skp1/Cullin/F-box (SCF) protein complex”

L79, when using the abbreviation APC/C, please also introduce the “cyclosome” in line 78.

We have now introduced the “cyclosome” as suggested by reviewer 3.

For example, L94 or L100, the complex serves pleiotropic function and implicitly restricting this function to the processes that were analysed is probably an underestimation.

We have now modified the sentences to indicate that the mentioned processes require the complex for their completion.

L117, Fig. 1A and B do not show a 10 but an 8-minute time point, please clarify. Also, A and B look very different and it would be worth harmonizing the labels, etc.

We thank reviewer 3 for spotting this discrepancy. We have corrected the main text and significantly reformatted Figure 1.

Fig. 1C Should the title not be “pre-treatment” as in the caption? Perhaps clearer would be “treatment pre-activation” and in Fig. 1D “treatment post-activation”.

We thank reviewer 3 for these suggestions that we have implemented.

Fig. 1D, the order of example images/categories and quantification thereof seems wrong, please reorder. Also, if 20 microtubule-containing cells per condition were analysed, showing the percentages of the different morphologies would be better. The presence of MTs is irrelevant, since these cells were chosen based on this.

This is another good point raised by reviewer 3. We now show the percentages of different morphologies in Figure 1C/D as observed by Giemsa staining. We have moved the U-ExM analysis in Fig. S1B-C to only mention that treated parasites were still assembling a network of subpellicular microtubules.

L129, please guide the reader with arrow heads to the apical complex in Fig. 1D.

We have added arrows to guide the readers to the apical pole.

L145, IMC and glideosome are not in the list of enriched GO terms and this sentence would benefit from providing names of proteins that were more detected and which have a clear link to these structures. Also, please reorder GO terms and the volcano plot in Fig. 1E, as the GO enrichment is based on the data presented in the volcano plot.

As suggested by reviewer 4, we have performed additional triplicates, which reduced the number of proteins differentially detected across the two time points. We thus only present a GO term enrichment analysis for ubiquitinated proteins in non-activated and 4 min activated gametocytes.

Fig. 1G, the “G” seems hidden in my copy. Please check.

We thank reviewer 3 for spotting this error. Figure 1 has been significantly reformatted and there is no more G panel.

L164, 166, and elsewhere, please harmonize the mentioning of 3xHA-tags.

We have harmonised the nomenclature as suggested by reviewer 3.

L166, again it is confusing, that data on FBXL2 and FBXO1 are shown in Fig. 2B although they are only identified as candidates later. Please revise. Also, the data on CUL1-HA is missing in Fig. 2B.

We thank reviewer 3 for this suggestion, we have made distinct panels to describe FBXL2 and FBXO1. CUL1-HA data was not missing, we are just not able to detect the protein by western blotting. As suggested by reviewers 1 and 2, we have moved this panel to Figure S2B and clearly indicate in the text, figure legend and method section that we are not able to detect the protein by western blotting.

L174, why would only some proteins of the complex re-localize as shown in Fig. 2B? Please discuss.

This is an interesting question for which we do not have experimental answers. Co-immunoprecipitations would only interrogate the protein populations interacting with each other. It is possible that the ratio of interacting protein vs non-interacting protein is different for each component of the system leading to different apparent signals.

Fig 2I, please explain in the caption what has been quantified. Cells per micro litre, field of view? Also, what is nd? Not done or non-detected?

Figure 2I is now Figure 5C and now shows quantification of Giemsa-stained ookinetes instead of expanded cells. We have added representative images of banana-shaped and retort ookinetes to better explain the quantified phenotypes. We have removed from the manuscript the analysis of expanded ookinetes that was less robust (ND was standing for not detected).

L277, please comment and discuss the differences seen when comparing SKP1 localization in 2B and 3B?

We agree we cannot observe obvious enrichment at the mitotic spindle and axonemes of SKP1 by IFA using widefield microscope. Such enrichment is however consistently observed in confocal sections following expansion. It is possible that, in this case, expansion may unmask the targeted epitope, as previously described in other systems (see for example PMID30559430).

L253, Reordering Fig. 3G and H would be more logical (first show that the protein can be depleted, then show the effect) and a reference to the data showing the downregulation upon auxin addition seems missing in the text.

We thank reviewer 3 for this suggestion. We have modified the figure accordingly, these panels are now Figure 2E and 2F.

Fig. 3I, how many cells were analysed in the two biological replicates? Please also provide statistics.

We are now indicating this information in Figure 5D.

L264, this statement does not reflect what is shown in 4D. Here, it looks like only 30% are still Ter119 positive. Please check and provide statistics.

We thank reviewer 3 for spotting this discrepancy. The last author made a mistake when homogenising the colours across panels. We have now corrected the figure (now Figure 4F). We also provide statistics.

L269/Fig. 4B, while the shown image is very clear, a quantification of the enclosed vs. non-enclosed cells would be helpful. Perhaps the Ter119-staining in Fig. 4A can provide that but this is currently unclear. Also, the centrin signal is not visible and an inset with the centrin plus another stain would fix the problem. Thus, the quantification of singlet vs. non-singlet centrin signal is unclear. In addition, the text states that segregation of centrosomes is delayed, hence, should the singles not precede the non-singlet? Please revise.

As indicated above the colours for Ter119-staining were inverted which made it difficult to understand the results. We have now corrected this mistake (new Figure 4F). We also highlight with black arrows the centrin-positive basal bodies (new Figure 4G).

During microgametogenesis, the eight basal bodies are rapidly formed de novo following activation of gametocytes by mosquito factors. During each following mitosis, sister basal bodies are segregated sequentially in the following order: 2 x 4 basal bodies, 4 x 2 basal bodies and 8 x 1 basal bodies. We are now indicating this in the result section.

L296, do you talk here about the gene or the protein? *fbxo1* vs. FBOX1?

We have changed *fbxo1* to FBOX1.

Fig. 5C, please explain what the “% of exflagellation centers per microgametocytes” is?

We thank reviewer 3 for highlighting this lack of clarity. We have modified the text in the result and method sections

Result section:

Disruption of *fbxo1* strongly impaired microgamete formation as determined by the percentage of microgametocytes leading to flagellated motile male gametes or exflagellation events (Fig. 4C).

Method section:

An exflagellation event was defined as moving flagellated parasites forming clumps (exflagellation centres) with nearby red blood cells.

Fig. 5D, as for Fig. 2I, please explain.

As for previous Fig. 2I, we have removed the previous panel 5D that was showing analysis from expanded parasites.

L341, Typo, “its” not “it”.

We thank reviewer 3 for spotting this typo. We have corrected it.

Fig. 6A, having the insets in a different orientation than the overview image makes it difficult to assess the signal in the bigger picture. Please revise. Similarly, in 6D. Please use the same frame and place a frame around the inset as well.

We have significantly reformatted Figure 6 which is now Figure 3 where insets are now in the same orientation as in the overview image.

L406/Fig. 6F, images exemplifying “normal” and “non-segregated nuclei” and “elongated and fragmented rhoptries” etc. are needed to assess the quantification of the phenotype in 6F. Please provide an image per category. If Fig. 6G should show these examples, please reorder the panels so the reader knows what has been quantified.

We now better described the phenotyping in the main text and show a corresponding annotated image for each category in Figure 3H. The text now reads:

FBXO1-depleted schizonts displayed less nuclei that were not fully segregated from each other (Fig. 3H-I and Fig. S3C). They also showed elongated and fragmented rhoptries which were not linked to an apical polar ring (APR) as seen by NHS-ester staining of expanded schizonts. The subpellicular microtubules were also misshaped, making it difficult to count them, and did not radiate evenly from an APR in FBXO1-depleted schizonts.

Fig. 6G, please provide where the insets have been cropped.

We have added this indication in all images.

L411, please provide a reference where the corresponding imaging data can be found. Currently it is unclear how uncoupled apical complexes and nuclei look like. Is this sentence based on the data shown in 6F? If so, it is unclear what “50% of observed apical complexes in FBXO1-depleted schizonts” mean. 50% comparing +IAA all vs mis.?

Please see our response to comment “L406/Fig. 6F”

Reviewer #4 (Remarks to the Author):

General Comments

In the manuscript NCOMMS-22-30599 “The Skp1-Cullin1-FBXO1 complex is a pleiotropic regulator required for the formation of gametes and zoites in *Plasmodium berghei*”, the authors examine ubiquitination in SCF complexes. Overall I thought this was a good manuscript, well written, well presented and supported. One concern was Figure 1 and labelling errors and omission of labels. However, my main concern is that it appears only 2 biological replicates were used for the majority of the proteomic experiments. 2 biological replicates are not acceptable for the statistical analyses derived from the experiments. Thus the significance of the data is suspect.

We thank reviewer 4 for these constructive comments and suggestions. We have reformatted Figure 1 and performed the proposed additional replicates as suggested in the specific comments.

Specific Comments

1. Figure 1 has mis-labels and discrepancies that make this section confusion and difficult to read. Line 117 – 119 describes figure 1A-B as anti-ubiquitin and anti-ubiquitin K-48. Figure 1A shows labels DMSO and 1 mM MG132 with s-UbR on the bottom, and 1B has labels anti-Ub K48 and anti UB-K48 for each. Please redo this figure with clear and consistent labeling. On line 881 there is mention of Figure 1G, but there is no G labeled on the Figure (I can assume which it is, but it should be labeled). And there is no label for the U-ExM images.

We thank reviewer 4 for also pointing to this mis-labels and discrepancies. We have taken these comments into account when preparing the revised set of figures.

2. Figure S1 is a supplementary Western, the Ponceau control is not shown. The quality of the Western is questionable.

We have added the Ponceau staining and additionally included blots with α -Ub K48 and α -Ub K63 staining showing similar patterns.

3. Line 877 regarding Figure 1E states 2 biological replicates were used. At minimum, 3 biological replicates is required to acquire meaningful statistics. I have major concerns about reporting these results with 2 biological replicates. Similar with line 948 regarding Figure 4E and line 959 Figure 5A where only two biological replicates are used, and Figure 5I/5K with only one biological replicate.

We thank reviewer 4 for raising these concerns. We are now providing biological triplicates for previous Figure 1E (now moved to Fig. S1D as suggested by reviewer 1), previous Figure 4E (now Fig. 4I), previous Figure 5I (now Fig. 6J). Results shown in previous Figure 5K (now Fig. 6G/H/I) were already from triplicates. These new replicates did not alter the main conclusions of the manuscript.

Results displayed in previous Figure 5A show reciprocal IPs with clear enrichment compared with duplicates IPs of 17 other proteins in gametocytes and a WT control (see response to reviewer 1). Therefore, we felt that adding a third replicate would not justify the use of additional animals.

4. Over all the methods are very detailed. Please double check line 731. Matrix Science calculates the emPAI value. I am not aware that Scaffold does emPAI, I could be mistaken so please confirm this detail.

Scaffold is indeed calculating emPAI values. The quantitative method can be selected in the “experiment > quantitative analysis section”.

5. The biological replicate information is only disclosed in the Figure legends. The number of biological replicates should be described in detail in the Methods.

We are now stating the number of replicates both in the method section and the figure legends whenever possible.

REVIEWER COMMENTS

Reviewer #1 (Remarks to the Author):

The revised manuscript is an enormous improvement on the original submission. The authors have satisfactorily addressed all the comments and they have produced a much more logical and cohesive narrative that presents a highly impactful and interesting story.

Although the diGly profiling of gametocytes pre and post activation doesn't reveal anything notable, the data is now presented in a way that does not detract from the focus of the paper.

The reorganisation makes the cdk1 data and the reasoning behind these experiments much easier to follow and does an excellent job of highlighting the major findings.

One very minor typographical error that I noticed was on line 302, '4 minutes upon stimulation by XA' should be changed to '4 minutes following stimulation by XA'.

I recommend approval for publication.

Reviewer #2 (Remarks to the Author):

The manuscript has improved from the original submission, but still lacks data to support the mechanistic claims of the phenotypes observed following FBXO1 deletion. I have outlined this in more detail below along with some other comments.

Major concerns

Fig. 3 - I am not convinced that FBXO1-HA is located at the IMC in schizonts as stated. Fluorescent images of FBXO1-HA and MTIP co-labelling have been supplied. While I agree that there are some regions of overlap in the MTIP labelling in late schizonts, there is little overlap in early schizonts. There appears to be a large amount of cytoplasmic staining in addition to some specific staining, but it is difficult to tell exactly where it is locating at this resolution.

The expansion images are nice and show FBXO1 accumulating on the cytoplasmic side of the centrosome forming a plate/structure that then appears to locate with the APR and rhoptries facing the exterior of the cell. But how exactly this relates to the IMC isn't clear as images to support the conclusions are not provided.

The early schizont image in Fig. 3D shows nice FBXO1 caps/plates which would be equivalent to what is seen in the images of Fig 3A and B, but there is no IMC colocalization at this stage. Expansion images of IMC labelling would be needed to see what is going on.

In the zoom image shown in figure 3C, it appears that FBXO1 is encasing what I assume is the NHS ester stained rhoptries based on the size and position? If it was the IMC moving down the merozoite as it forms I would expect it to be a larger ring. The other examples in this image show tight association of FBXO1 with the APR and rhoptries, but without the pronounced ring. As the final step of cytokinesis has been reported to be synchronous, would you not expect a similar pattern across the cell?

The tight association of FBXO1 with the APR/rhoptries appears to be maintained during mitosis, but then lost in segmented schizonts with some weaker non-specific labelling (as per figure S3A right panel). This labelling doesn't look like IMC, which you would expect to be encasing the formed merozoites as depicted in your schematic. Instead, the labelling is sparse and looks non-specific in the segmented schizont.

Fig 4. In the version of the figures I have it is impossible to see the centrin staining in part G. It would be good to see the centrin staining more clearly as the proposed defect relates to centrosome positioning.

Figure 5 and related section.

The conclusion that the defect in development lies in IMC formation is not supported here. The cell shape is altered, but in Fig S5B you show that there is CDPK1-HA staining (used as an IMC marker) at the periphery of the ookinete in addition to some internal staining. This suggests that the IMC has almost completely formed as it is almost around the whole periphery.

In Fig 5E the subpellicular microtubules are at the periphery of the cell but you do not see the regular spaced MT network, which may suggest a defect in aligning the MTs to the IMC. The microtubules are responsible for elongating the cell, so in this situation if they are not anchored to the IMC then the shape may get distorted.

In addition to the IMC related proteins within the ookinete IPs there also appears to be enrichment of alpha and beta tubulin. It should also be noted that the IP experiments were performed on cells that had been fixed with formaldehyde, which would crosslink proteins. As the IMC and MT networks are so closely positioned you may be pulling down proteins that are both direct interacting proteins and more distant ones that are part of larger complexes.

Do you think the defect may lie in linking the IMC and MT networks rather than in IMC formation perse?

Other comments

Line 123 – fix fig reference (Fig. 1C-D) no E

Fig S1 D and E –Some of the colours have not reproduced well or at all. A higher resolution image should be included in the final version. Also in the figure legend there is reference to “The panel on the left shows GO term enrichment analysis for down- and up regulated proteins”. This needs to be removed as it not part of the figure anymore.

Line 163 – Re Cul1-HA detection - perhaps add a comment here similar to what you have put in the methods or refer to the methods section here.

Line 181 - IFA showed a similar cytoplasmic distribution of FBXO1-HA and FBXL2-HA in both macro- and microgametocytes (Fig. 2E). You should mention the brighter punctate staining in the microgametes as these are absent in the RBX1/SKP1.

Fig. 2H – it is difficult to see any signal in the RBX1-HA schizont in my version of the figures.

Line 198/fig 2H and I – RBX1-appears to be very weak in schizonts and completely missing in ookinets? What do you think is going on here? As it is a part of the SCF core does this mean the complex doesn't form/function in these stages?

Fig. 2J – doesn't appear to be referenced in the text. RBX1 should be added to this as well. in the text here when you say expression do you mean protein?

General comment - Some of the images are still difficult to see. Multiple layers sometimes detract/mask the staining pattern. It is difficult to see the green on top of the greyscale NHS staining. Some of these issues may be due to the quality of the PDF which appeared to be pixilated.

Reviewer #3 (Remarks to the Author):

The revised version of the manuscript addresses my previous comments and it gained more clarity, which was one of my major concerns. The narrative is much clearer and the manuscript is more assessable, in particular to the non-expert reader.

Minor comments:

Fig. 4A, B / Fig. 6 B: perhaps consider highlighting the zoom-in in the actual merged image rather than the single channel image. Like it has been nicely done in Fig. 5A.

One thing which I could not find in the Methods section is the description of how the data in the Fig. 2G were generated. Please check.

Please consistently label CDPK1^K62A, the 'A' is occasionally missing.

Reviewer #4 (Remarks to the Author):

This is a much stronger and clearer manuscript. The new and improved figures are a huge improvement to the quality of the paper. All of my concerns have been more than adequately addressed.

REVIEWER COMMENTS

Reviewer #1 (Remarks to the Author):

The revised manuscript is an enormous improvement on the original submission. The authors have satisfactorily addressed all the comments and they have produced a much more logical and cohesive narrative that presents a highly impactful and interesting story.

Although the diGly profiling of gametocytes pre and post activation doesn't reveal anything notable, the data is now presented in a way that does not detract from the focus of the paper.

The reorganisation makes the cdk1 data and the reasoning behind these experiments much easier to follow and does an excellent job of highlighting the major findings.

One very minor typographical error that I noticed was on line 302, '4 minutes upon stimulation by XA' should be changed to '4 minutes following stimulation by XA'.

I recommend approval for publication.

We thank reviewer 1 for the positive feedback.

We have corrected the typographical error.

Reviewer #2 (Remarks to the Author):

The manuscript has improved from the original submission, but still lacks data to support the mechanistic claims of the phenotypes observed following FBXO1 deletion. I have outlined this in more detail below along with some other comments.

We thank reviewer 2 for their positive feedback and for further suggestions to improve the clarity of our manuscript.

Major concerns

Fig. 3 - I am not convinced that FBXO1-HA is located at the IMC in schizonts as stated. Fluorescent images of FBXO1-HA and MTIP co-labelling have been supplied. While I agree that there are some regions of overlap in the MTIP labelling in late schizonts, there is little overlap in early schizonts. There appears to be a large amount of cytoplasmic staining in addition to some specific staining, but it is difficult to tell exactly where it is locating at this resolution.

We indeed do not have direct evidence at the correct resolution to claim IMC localisation in merozoites. We have reworded statements that may have been confusing.

We have changed:

“In contrast, SKP1-HA, FBXO1-HA, and FBXL2-HA showed an apical localisation that was reminiscent of the IMC localisation of *P. falciparum* GAP45 in late schizogony [34] (Fig. 2H).”

to:

“In contrast, SKP1-HA, FBXO1-HA, and FBXL2-HA showed a pattern that was reminiscent of the dynamic localisation of the glideosome-associated protein 45 (GAP45) during *P. falciparum* late schizogony [34] (Fig. 2H).”

The expansion images are nice and show FBXO1 accumulating on the cytoplasmic side of the centrosome forming a plate/structure that then appears to locate with the APR and rhoptries facing the exterior of the cell. But how exactly this relates to the IMC isn't clear as images to support the conclusions are not provided.

We thank reviewer 2 for pointing out that our wording may have been confusing.

Our EM images indeed show that IMC is formed. However, merozoite segmentation is defective. We have now reworded the discussion to avoid any confusion as follow:

“The observed phenotypes suggest that FBXO1 is important in the following steps for proper biogenesis of the apical complex ~~and IMC formation~~, as observed for the basal complex protein PfcINCH [59].”

to:

“The observed phenotypes suggest that FBXO1 is important in the steps following proper ~~assembly~~ of the apical complex ~~and merozoite segmentation~~, as observed for the basal complex protein PfcINCH [59].”

The early schizont image in Fig. 3D shows nice FBXO1 caps/plates which would be equivalent to what

is seen in the images of Fig 3A and B, but there is no IMC colocalisation at this stage. Expansion images of IMC labelling would be needed to see what is going on.

We agree with reviewer 2 that a further characterisation of the relationship between FBXO1 localisation and the IMC would be of interest. However, as we have modified our wording suggesting a direct functional relationship between FBXO1 and IMC formation, we think that detailed U-ExM analysis of IMC formation upon depletion of FBXO1 would represent an experiment for a distinct follow-up manuscript.

In the zoom image shown in figure 3C, it appears that FBXO1 is encasing what I assume is the NHS ester stained rhoptries based on the size and position? If it was the IMC moving down the merozoite as it forms I would expect it to be a larger ring. The other examples in this image show tight association of FBXO1 with the ARP and rhoptries, but without the pronounced ring. As the final step of cytokinesis has been reported to be synchronous, would you not expect a similar pattern across the cell?

We agree with reviewer 2 that most of the final cell division events show a similar pattern across the schizont as seen in Fig. 2H, 3A-D, S3A.

The tight association of FBXO1 with the APR/rhoptries appears to be maintained during mitosis, but then lost in segmented schizonts with some weaker non-specific labelling (as per figure S3A right panel). This labelling doesn't look like IMC, which you would expect to be encasing the formed merozoites as depicted in your schematic. Instead, the labelling is sparse and looks non-specific in the segmented schizont.

We thank reviewer 2 for pointing out that the full projections we provided did not highlight the peripheral localisation of FBXO1-HA in merozoites. We have now added a panel in Fig. S3 showing a confocal section. This highlights the dotted peripheral localisation we mention in the text and in our model.

Fig 4. In the version of the figures I have it is impossible to see the centrin staining in part G. It would be good to see the centrin staining more clearly as the proposed defect relates to centrosome positioning.

We agree with reviewer 2 that the images in the main figure are small. We have added more images in panel C of Fig. S4 to highlight the centrin staining.

Figure 5 and related section.

The conclusion that the defect in development lies in IMC formation is not supported here. The cell shape is altered, but in Fig S5B you show that there is CDPK1-HA staining (used as an IMC marker) at the periphery of the ookinete in addition to some internal staining. This suggests that the IMC has almost completely formed as it is almost around the whole periphery.

In Fig 5E the subpellicular microtubules are at the periphery of the cell but you do not see the regular spaced MT network, which may suggest a defect in aligning the MTs to the IMC. The microtubules are responsible for elongating the cell, so in this situation if they are not anchored to the IMC then the shape may get distorted.

We agree we still observe parts of IMC by EM and the use of “formation” was indeed misleading. We are now using “integrity” to avoid any confusion. We paid attention to not call CDPK1 an IMC marker but a pellicular marker, as its exact localisation has not been assessed yet. We have rephrased our conclusion 1338-340 to: “Altogether these results indicate a role of FBXO1 in maintaining the integrity of the pellicle and the subpellicular microtubule network during ookinete development. We have also changed the title of the section accordingly”.

It was recently published that ookinetes lacking certain apical polar ring proteins show a disconnection between the IMC and the subpellicular microtubules. This leads to fully formed ookinetes that however do not display the characteristic bent banana-shaped ookinete. Here the phenotype seems more pronounced as the development is arrested at the retort stage.

In addition to the IMC related proteins within the ookinete IPs there also appears to be enrichment of alpha and beta tubulin. It should also be noted that the IP experiments were performed on cells that had been fixed with formaldehyde, which would crosslink proteins. As the IMC and MT networks are so closely positioned you may be pulling down proteins that are both direct interacting proteins and more distant ones that are part of larger complexes.

Do you think the defect may lie in linking the IMC and MT networks rather than in IMC formation perse?

We agree with reviewer 2 that the subpellicular microtubule network is likely not properly anchored to the IMC as observed in Fig. 5E, at least due to the loss of integrity of the IMC observed in Fig. 5F. However, we still observe by EM subpellicular microtubules that are closely apposed to IMC sections as seen in Fig. 5G. This would require more detailed three-dimensional quantification, which, we feel, would represent an experiment for a distinct follow-up project.

Other comments

Line 123 – fix fig reference (Fig. 1C-D) no E

We thank reviewer 2 for spotting this discrepancy. We have corrected the text.

Fig S1 D and E –Some of the colours have not reproduced well or at all. A higher resolution image should be included in the final version. Also in the figure legend there is reference to “The panel on the left shows GO term enrichment analysis for down- and up regulated proteins”. This needs to be removed as it not part of the figure anymore.

We have now included higher resolution images.

We thank reviewer 2 for spotting this inconsistency between the figure legend and the figure. We have removed the sentence.

Line 163 – Re Cul1-HA detection - perhaps add a comment here similar to what you have put in the methods or refer to the methods section here.

We have modified the text as follows:

Immunoblotting confirmed expression of fusion proteins in gametocytes with the expected mobility for RBX1-HA and SKP1-HA (Fig. 2A) but no signal above the background could be observed for CUL1-HA (Fig. S2B) from two independent lines despite correct in-frame integration of the HA tag (see method section).

Line 181 - IFA showed a similar cytoplasmic distribution of FBXO1-HA and FBXL2-HA in both macro- and microgametocytes (Fig. 2E). You should mention the brighter punctate staining in the microgametes as these are absent in the RBX1/SKP1.

We have modified the text as follows:

IFA showed a similar cytoplasmic distribution of FBXO1-HA and FBXL2-HA in both macro- and microgametes with slightly brighter staining at structures that likely correspond to mitotic spindles and forming axonemes in the microgametes (Fig. 2E).

Fig. 2H – it is difficult to see any signal in the RBX1-HA schizont in my version of the figures.

The signal is indeed very low. We have now uploaded the original image without compression, which we hope will allow to better visualise the faint signal.

Line 198/fig 2H and I – RBX1-appears to be very weak in schizonts and completely missing in ookinetes? What do you think is going on here? As it is a part of the SCF core does this mean the complex doesn't form/function in these stages?

We did not directly investigate RBX1 expression in ookinetes. Previous transcriptomic analyses showed similar *rbx1* transcript levels across the schizont, gametocyte and ookinete stages (PMID28081440). As RBX1, in comparison to SKP1, FBXO1 and FBXL2, is less well detected by MS and IFA in both schizonts and gametocytes, we cannot rule out RBX1 expression and incorporation in the complex at the ookinete stage. It is of course a possibility that other functionally related proteins may be part of the complex. It is also possible that other proteins with similar function are incorporated in the complex. However, we feel we do not have data currently supporting these hypothesis and prefer to not discuss them.

Fig. 2J – doesn't appear to be referenced in the text. RBX1 should be added to this as well. in the text here when you say expression do you mean protein?

We thank reviewer 2 for spotting these omissions. We are now referencing the figure in the text and indicate "protein expression".

General comment - Some of the images are still difficult to see. Multiple layers sometimes detract/mask the staining pattern. It is difficult to see the green on top of the greyscale NHS staining. Some of these issues may be due to the quality of the PDF which appeared to be pixilated.

We are now providing less compressed images and show inset for the green centrin staining in FBXO1-GD gametocytes.

Reviewer #3 (Remarks to the Author):

The revised version of the manuscript addresses my previous comments and it gained more clarity, which was one of my major concerns. The narrative is much clearer and the manuscript is more assessable, in particular to the non-expert reader.

Minor comments:

Fig. 4A, B / Fig. 6 B: perhaps consider highlighting the zoom-in in the actual merged image rather than the single channel image. Like it has been nicely done in Fig. 5A.

We thank reviewer 3 for this suggestion that we have implemented.

One thing which I could not find in the Methods section is the description of how the data in the Fig. 2G were generated. Please check.

We thank reviewer 3 for spotting this omission. The protein-protein interaction network of enriched proteins was represented using Cytoscape 3.9.0. We now indicate this in the method section.

Please consistently label CDPK1^{K62A}, the 'A' is occasionally missing.

To avoid any confusion, we now indicate CDPK1-K62 to indicate lysine 62 of CDPK1 and CDPK1^{K62A} to indicate the substitution.

Reviewer #4 (Remarks to the Author):

This is a much stronger and clearer manuscript. The new and improved figures are a huge improvement to the quality of the paper. All of my concerns have been more than adequately addressed.

We thank again reviewer 4 for their constructive and supportive comments.